**Aerosol Responses to Precipitation Along North American Air Trajectories Arriving at Bermuda**

Hossein Dadashazar[1], Majid Alipanah[2], Miguel Ricardo A. Hilario[3], Ewan Crosbie[4,5], Simon Kirschler[6,7], Hongyu Liu[8], Richard H. Moore[4], Andrew J. Peters[9], Amy Jo Scarino[4,5], Michael Shook[4], K. Lee Thornhill[4], Christiane Voigt[6,7], Hailong Wang[10], Edward Winstead[4,5], Bo Zhang[8], Luke Ziemba[4], Armin Sorooshian[1,3]

[1]Department of Chemical and Environmental Engineering, University of Arizona, Tucson, AZ, USA
[2]Department of Systems and Industrial Engineering, University of Arizona, Tucson, AZ, USA
[3]Department of Hydrology and Atmospheric Sciences, University of Arizona, Tucson, AZ, USA
[4]NASA Langley Research Center, Hampton, VA, USA
[5]Science Systems and Applications, Inc., Hampton, VA, USA
[6]Institute for Atmospheric Physics, DLR, German Aerospace Center, Oberpfaffenhofen, Germany
[7]Institute for Atmospheric Physics, University of Mainz, Mainz, Germany
[8]National Institute of Aerospace, Hampton, VA, USA
[9]Bermuda Institute of Ocean Sciences, 17 Biological Station, St. George's, GE01, Bermuda
[10]Atmospheric Sciences and Global Change Division, Pacific Northwest National Laboratory, Richland, WA, USA

*Correspondence to: Hossein Dadashazar (hosseind@arizona.edu)

**Abstract**

North American pollution outflow is ubiquitous over the western North Atlantic Ocean, especially in winter, making this location a suitable natural laboratory for investigating the impact of precipitation on aerosol particles along air mass trajectories. We take advantage of observational data collected at Bermuda to seasonally assess the sensitivity of aerosol mass concentrations and volume size distributions to accumulated precipitation along trajectories (APT). The mass concentration of particulate matter with aerodynamic diameter less than 2.5 μm normalized by the enhancement of carbon monoxide above background ($PM_{2.5}/\Delta CO$) at Bermuda was used to estimate the degree of aerosol loss during transport to Bermuda. Results for December-February (DJF) show most trajectories come from North America and have the highest APTs, resulting in significant reduction (by 53%) in $PM_{2.5}/\Delta CO$ under high APT conditions (> 13.5 mm) relative to low APT conditions (< 0.9 mm). Moreover, $PM_{2.5}/\Delta CO$ was most sensitive to increases in APT up to 5 mm (-0.044 $\mu g\ m^{-3}\ ppbv^{-1}\ mm^{-1}$) and less sensitive to increases in APT over 5 mm. While anthropogenic $PM_{2.5}$ constituents (e.g., black carbon, sulfate, organic carbon) decrease with high APT, sea salt in contrast was comparable between high and low APT conditions owing to enhanced local wind and sea salt emissions in high APT conditions. The greater sensitivity of the fine mode volume concentrations (versus coarse mode) to wet scavenging is evident from AERONET volume size distribution data. A combination of GEOS-Chem model simulations of $^{210}Pb$ submicron aerosol tracer and its gaseous precursor $^{222}Rn$ reveal that (i) surface aerosol particles at Bermuda are most impacted by wet scavenging in winter/spring (due to large-scale precipitation) with a maximum in March, whereas convective scavenging plays a substantial role in summer; and (ii) North American $^{222}Rn$ tracer emissions contribute most to surface $^{210}Pb$ concentrations at Bermuda in winter (~75-80%), indicating that air masses arriving at Bermuda experience large-scale precipitation scavenging while traveling from North America. A case study flight from the ACTIVATE field campaign on 22 February 2020 reveals a significant reduction in aerosol number and volume concentrations during air mass transport off the U.S. East Coast associated with increased cloud fraction and precipitation. These results highlight the sensitivity of remote marine boundary layer aerosol characteristics to precipitation along trajectories, especially when the air mass source is continental outflow from polluted regions like the U.S. East Coast.

## 1. Introduction

Aerosol properties are difficult to characterize in remote marine regions owing to the scarcity of monitoring stations as compared to over land. Island observatories are critical resources to investigate long-range transport of aerosol particles and their associated properties (e.g., Silva et al., 2020). The western North Atlantic Ocean (WNAO) includes the island of Bermuda, which has a rich history of monitoring data for both surface and columnar aerosol characteristics, thus affording the opportunity to study how aerosol properties are impacted by different sources and processes along the transport of air masses to the site. Consequently, Bermuda has been the subject of decades of intense atmospheric science research (Sorooshian et al., 2020), as it is a receptor site for both North African dust (Chen and Duce, 1983) and anthropogenic outflow from both North America (Arimoto et al., 1992; Galloway et al., 1989; Moody et al., 2014; Corral et al., 2021) and Europe (Anderson et al., 1996; Cutter, 1993). North American outflow reaching Bermuda has been linked to appreciable levels of anthropogenic species (e.g., sulfate, lead, elemental carbon, ozone) (Wolff et al., 1986), more acidic rainfall as compared to other air mass sources (Jickells et al., 1982), and a significant reduction of sulfate levels in both aerosol and wet deposition samples in response to reduced $SO_2$ emissions in recent decades (Keene et al., 2014).

There have been extensive studies reporting on some aspect of air mass history, normally by calculating air parcel trajectories using transport and dispersion models, prior to arrival at Bermuda (Sorooshian et al., 2020 and references therein), including predominant circulation patterns impacting Bermuda at different times of the year (e.g., Miller and Harris, 1985; Veron et al., 1992). What remains uncertain is how precipitation along those trajectories impacts surface aerosol characteristics at Bermuda. Wet scavenging rates are very difficult to constrain over open ocean areas such as the WNAO (Kadko and Prospero, 2011) not only because of complexity of physical mechanisms in play but also scarce necessary field measurements. Overall, more work is warranted to better constrain wet scavenging of aerosol particles along trajectories as such studies are sparse not only for the WNAO but also for other regions (Tunved et al., 2013; Hilario et al., 2021). Arimoto et al. (1999) used aerosol radionuclide data in relation to airflow pattern information to conclude that pollutant transport to Bermuda is common from the northwest and that precipitation scavenging can be influential; their analysis of rain effects on nuclide activities were based on rain data collected at Bermuda without knowledge of precipitation transport history prior to arrival. While many studies have investigated how composition at Bermuda varies based on air mass trajectories (Miller and Harris, 1985; Cutter, 1993; Huang et al., 1996), the subject of how precipitation along those trajectories impact the resultant aerosol at Bermuda has not been adequately addressed but is motivated by past works (Moody and Galloway, 1988; Todd et al., 2003).

In their recent aerosol climatology study for Bermuda, Aldhaif et al. (2021) found the peculiar result that fine particulate pollution in the winter months (December-February) was reduced even though there was an enhanced number density of air mass back trajectories traced back to North America. They hypothesized that enhanced seasonal cloud fractions and precipitation in winter (Painemal et al., 2021) contribute to the removal of aerosol particles during transport via wet scavenging, which we aim to study more deeply here using a variety of datasets. Results of this study have broad relevance to all remote marine regions impacted by transported continental pollution, in addition to advancing knowledge of how precipitation can impact surface aerosol characteristics.

## 2. Datasets and Methods

Datasets used in this work are summarized in Table 1 and described in brief detail below.
**Table 1. Summary of datasets used in this work. Data are between 1 January 2015 and 31**
**December 2019, with the exception of ACTIVATE aircraft data based on a single flight day**
**on 22 February 2020. Section 2 provides more details about the datasets used in this study,**
**including specific instruments from the ACTIVATE airborne dataset.**

| Parameter | Acronym | Data Source | Spatial Resolution | Time Resolution |
|---|---|---|---|---|
| Particulate matter mass concentration (aerodynamic diameter less than 2.5 µm) | $PM_{2.5}$ | Fort Prospect Station | - | Hourly |
| Particulate matter mass concentration (aerodynamic diameter less than 10 µm) | $PM_{10}$ | Fort Prospect Station | - | Daily |
| Nitrogen monoxide concentration | NO | Fort Prospect Station | - | Hourly |
| Nitrogen dioxide concentration | $NO_2$ | Fort Prospect Station | - | Hourly |
| Nitrogen oxide concentration | $NO_X$ | Fort Prospect Station | - | Hourly |
| Volume size distribution | VSD | AERONET | - | Hourly |
| Carbon monoxide surface concentration | CO | MERRA-2 | $0.625° \times 0.5°$ | Hourly |
| Aerosol speciated surface mass concentrations | - | MERRA-2 | $0.625° \times 0.5°$ | Hourly |
| Surface wind speed | $Wind_{SF}$ | MERRA-2 | $0.625° \times 0.5°$ | Hourly |
| Planetary boundary layer height | PBLH | MERRA-2 | $0.625° \times 0.5°$ | Hourly |
| Precipitation | APT/Rain | GDAS | $1° \times 1°$ | Hourly |
| Aerosol/cloud properties | - | Airborne: ACTIVATE | - | 1 – 45 Sec |

**2.1 Bermuda Surface Measurements**
Aerosol and gas measurements were conducted at Fort Prospect in Bermuda (32.30° N,
64.77°W, 63 m ASL). Hourly $PM_{2.5}$ data were collected with a Thermo Scientific TEOM 1400a
Ambient Particulate Monitor with 8500C FDMS (Federal Equivalent Method EQPM-0609-181
for $PM_{2.5}$). Concentrations were determined by employing conditioned filter sample collection and
direct mass measurements using an inertial micro-balance (TEOM 1400a). Hourly precision was
$\pm 1.5 \mu g\ m^{-3}$. Hourly data were averaged over 6 hour intervals to match the time frequency of the
trajectory data discussed subsequently. The conversion of hourly data to 6 hour data also helps to
mask, to some extent, the unwanted effects of local sources and processes that occur on a small
timescale.
$PM_{10}$ concentrations were determined based on U.S. Environmental Protection Agency
(EPA) method IO-2 (EPA, 1999) using a Tisch model TE6070 hi-volume air sampler, equipped
with 8" × 10" TissuQuartz 2500 QAT-UP quartz fiber filters. The $PM_{10}$ sampler was operated at a
flow rate of 2.1 $m^3\ min^{-1}$ yielding a total volume of 3000 $m^3$ over a 24 hr sampling period. The
sampler flow rate was calibrated every 3 months. Sampling was synchronized with the 1-in-6 day
national ambient air quality schedule used by EPA. Prior to deployment, the filters were
equilibrated for 24 hr in an environmental control chamber maintaining constant conditions of
relative humidity (35 ± 2%) and temperature (21 ± 2°C). The filters were then weighed with a
precision of ± 0.1 mg using a Mettler Toledo AB104 balance, which was modified for weighing
unfolded 8" × 10" filters, and then transferred to clean re-sealable plastic bags for transportation to

the field site. After sampling, the exposed filters were returned immediately to the laboratory where they were re-equilibrated in the environmental control chamber for 24 hr before being re-weighed to determine the particle loading from which particle concentrations were calculated. $PM_{10}$ determinations have an accuracy of within $\pm$ 2.5%, which is equivalent to $\pm$ 0.2 µg m$^{-3}$ based on the average of $PM_{2.5}$ between 2015 and 2019 (i.e., 6.7 µg m$^{-3}$).

Various gases were monitored with hourly time resolution using a Model T200U Trace-level $NO/NO_2/NO_x$ analyzer (Teledyne API), which is a U.S. EPA compliance analyzer relying on a proven chemiluminescence principle. The gas analyzer was routinely calibrated using NIST-certified calibrant $NO_2$ in ultra-high purity nitrogen (Airgas, Inc., Radnor Township, PA, USA). Acceptable criteria applied for single point quality control (QC) allows for $\pm15.1\%$ or $< \pm1.5$ ppb difference, whichever is greater (40 CFR Part 58 App A Sec. 3.1.1). Similar to $PM_{2.5}$, these hourly gas data were averaged to 6-hour resolution.

There were a few periods when data were missing with the longest one being between 11 January 2016 and 08 April 2016 for the gases, and also between 16 October 2017 and 20 January 2018 for $PM_{2.5}$. There was no major discontinuity in $PM_{10}$ sampling. Table S1 reports the number of data points available for various seasons from the surface measurements at Fort Prospect in Bermuda.

Columnar aerosol data were obtained from a NASA AErosol RObotic NETwork (AERONET) (Holben et al., 1998) surface station at Tudor Hill (32.264° N, 64.879° W). Level 2 daily data have been quality assured and cloud screened based on the Version 3 algorithm (Giles et al., 2019). We focus on the volume size distribution (VSD) product that has 22 logarithmically equidistant discrete radii ranging from 0.05 to 15 µm. A radius of 0.6 µm typically discriminates between fine and coarse modes when using AERONET data (Dubovik et al., 2002; Schuster et al., 2006).

## 2.2 Reanalysis Data

Modern-Era Retrospective analysis for Research and Applications-Version 2 (MERRA-2) (Gelaro et al., 2017) products were used as a data source for speciated aerosol and gas parameters including surface mass concentration of sea-salt (collection "tavg1_2d_aer_Nx") and surface concentration of carbon monoxide (CO; collection "tavg1_2d_chm_Nx"). Surface wind speed and planetary boundary layer height (PBLH) (collection "tavg1_2d_flx_Nx") data were also obtained from MERRA-2. Hourly and 3-hourly data were downloaded and averaged for a 0.5° latitude by 0.625° longitude grid (i.e., 32° – 32.5°N and 64.375° – 65°W) surrounding Bermuda and subsequently averaged over 6-hour intervals to match the time frequency of trajectory analysis results. It should be noted that MERRA-2 data were temporally and spatially coincident with the ending point of trajectories over Bermuda. The Global Data Assimilation System (GDAS) one-degree archive data were used for trajectory calculations explained in the subsequent section. Precipitation data were also obtained along the trajectories based on GDAS one-degree data.

## 2.3 Air Mass Trajectory Analysis

To track air mass pathways arriving at Bermuda (32.30° N, 64.77°W), we obtained 10-day (240 hr) back-trajectories from the Hybrid Single-Particle Lagrangian Integrated Trajectory model (HYSPLIT) (Stein et al., 2015; Rolph et al., 2017). We used an ending altitude of 100 m (AGL) to be within the surface layer and close to the measurement site. As discussed later, sensitivity analysis with higher ending altitudes (500 m and 1 km; Figs. S1-S2) reveals similar results to using 100 m. Four trajectories were initialized (i.e., 6-hour interval) each day between 1 January 2015

00:00:00 UTC and 31 December 2019 18:00:00 UTC resulting in a total of 7304 individual
trajectories. Trajectories were calculated using the GDAS one-degree archive data and with the
"model vertical velocity" method, which means vertical motions were handled directly using
meteorological data files. Moreover, accumulated precipitation along trajectories (APT) was
calculated by integrating surface precipitation data from GDAS throughout the transport to the
receptor site. As GDAS precipitation data corresponds to the surface level, it should be noted that
APT values presented in this study are associated with the potential maximum level of
precipitation experienced by the air parcel through its transport journey. Results presented in Figs.
1-3 are based on 10-day back-trajectories, whereas analyses presented in the remaining sections of
the paper are based on 4-day (96 hr) back-trajectories.
Trajectory analyses contain errors that originate from factors including, but not limited to,
the choice of input meteorological data, resolution of input data, and the vertical transport method
used in trajectory calculations (Stohl et al., 1995; Cabello et al., 2008; Engström and Magnusson
2009). Although the choice of meteorological data is the most important contributor to the
uncertainties associated with trajectories calculations (Gebhart et al., 2005), no particular dataset
has been found to be superior in terms of yielding the lowest error. While in this study we used
GDAS data, which have been widely used as input dataset for trajectory calculations even in
regions with complicated topography (e.g., Tunved et al., 2013; Su et al., 2015), the
aforementioned inherent errors should not be overlooked when interpreting the results presented
in this work. Another factor that can contribute to the uncertainties for the results presented in this
work is the use of GDAS as the source of precipitation data as previous works (Sun et al., 2018;
Nogueira 2020) have demonstrated that there is some level of disagreement between precipitation
datasets.
**2.3.1 Concentration Weighted Trajectory Analysis and Seasonal Rain Maps**
Concentration weighted trajectories (CWT) were calculated based on the 10-day back-
trajectories from HYSPLIT in conjunction with Bermuda surface $PM_{2.5}$ data described in Section
2.1. The CWT method has been implemented widely to identify long-range pollutant transport
pathways impacting a receptor site (Hsu et al., 2003; Wang et al., 2009; Hilario et al., 2020).
Seasonal maps of average precipitation experienced by trajectories were also estimated based on
10-day back-trajectories from HYSPLIT. The aforementioned analyses were performed for
$0.5°\times0.5°$ grids covering the area encompassed by 10°–80°N and 5°–170°W. A weight function
($W_{ij}$ in Eq. 1) following the method of Dimitriou et al. (2015) was applied in the CWT analysis
and precipitation maps to increase statistical stability. In Eq. 1, $n_{avg}$ is the average number of
trajectory end points per individual gird cell over the study region excluding cells with zero
trajectory points and $n_{ij}$ is the number of trajectory end points that lies in the grid cell (i,j).

$$
W_{ij} = \begin{cases} 1 & n_{ij} > 3\,n_{avg} \\ 0.7 & 1.5\,n_{avg} < n_{ij} < 3\,n_{avg} \\ 0.4 & n_{avg} < n_{ij} < 1.5\,n_{avg} \\ 0.2 & n_{ij} < n_{avg} \end{cases} \tag{1}
$$


10-day back trajectories were implemented for generating CWT and rain maps to illustrate
potential distant sources impacting Bermuda. But for more quantitative analyses presented in the
subsequent sections focused on transport most relevant to the WNAO region, four-day back
trajectories were used by simply truncating 10-day trajectories. The use of four-day trajectories
reduces the uncertainties associated with trajectory calculations in comparison to using 10-day
trajectories and also enables us to focus on sources closer to the receptor site.
**2.3.2 Trajectory Clustering**
Hierarchical agglomerative clustering was used to identify characteristic trajectories
reaching Bermuda at 100 m (AGL). Hierarchical clustering was based on the "complete linkage"
method (Govender and Sivakumar, 2020). Four-day HYSPLIT back-trajectories were used to
perform clustering analysis. Distances between trajectories were calculated using the Haversine
formula, which calculates distance between two points on Earth assuming they are on a great circle
(Sinnott, 1984). The distance between any two trajectories was calculated as the sum of distances
between trajectory endpoints. Subsequently, clustering was conducted based on the symmetric
distance matrix, which includes the distances between all pairs of trajectories. Clustering was
performed for varying numbers of clusters, ranging between 2 and 32. The L-method (Kassomenos
et al., 2010) was implemented to identify the optimum number of clusters. In this method, root
mean square deviation (RMSD) was calculated for each clustering run and then plotted versus the
number of clusters to determine the optimum solution. RMSDs were estimated based on the
distances between trajectories and associated mean cluster trajectories.
**2.4 Airborne Measurements**
Airborne data from the Aerosol Cloud meteorology Interactions oVer the western ATlantic
Experiment (ACTIVATE) are used from Research Flight 6 (RF6) on 22 February 2020.
ACTIVATE involves two NASA Langley aircraft (HU-25 Falcon and UC-12 King Air) flying in
coordination at different altitudes to simultaneously characterize the same vertical column with a
focus on aerosol-cloud-meteorology interactions (Sorooshian et al., 2019). RF6 was a rare case of
the HU-25 Falcon flying alone, but this aircraft conveniently included measurements relevant to
this study. The ACTIVATE strategy involves the HU-25 Falcon flying in the boundary layer to
characterize gas, aerosol, cloud, and meteorological parameters along the following level legs:
Min. Alt. = lowest altitude flown (500 ft), BCB = below cloud base, ACB = above cloud base,
BCT = below cloud top, ACT = above cloud top.
Data from the following instruments were used: Condensation Particle Counter (CPC; TSI
Model 3772) for number concentration of particles with diameter > 10 nm; Scanning Mobility
Particle Sizer (SMPS; TSI Model 3081) for aerosol size distribution data between 3.2 – 89.1 nm;
Laser Aerosol Spectrometer (LAS; TSI Model 3340) for aerosol size distribution data between
diameters of 0.09 – 5 µm; two-dimensional optical array imaging probe (2DS; SPEC Inc.) (Lawson
et al., 2006) for rain water content (RWC) quantified by integrating rain drop size distributions
between diameters of 39.9 – 1464.9 µm; and Fast Cloud Droplet Probe (FCDP; SPEC Inc.) (Knop
et al., 2021) for cloud liquid water content (LWC) calculated by integrating drop size distributions
between diameters of 3 – 50 µm. With the exception of SMPS data (45 second resolution), all
airborne data were at 1 second resolution.
**2.5 Radionuclide tracers in GEOS-Chem Model**
Lead-210 ($^{210}$Pb, half-life 22.3 years) is the decay daughter of Radon-222 ($^{222}$Rn, half-life
3.8 days) emitted mainly from land surfaces. After production, it indiscriminately attaches to

ambient submicron particles, which move with the air until being scavenged by precipitation or deposited to the surface. Because of its relatively well-known source and wet deposition as its principal sink, $^{210}$Pb has long been used to test wet deposition processes in global models (e.g., Liu et al., 2001). It is also a useful tracer to describe continental air influence over oceans. In this study, we use $^{210}$Pb as simulated by the GEOS-Chem model to investigate the role of precipitation scavenging in affecting seasonal surface aerosol concentrations at Bermuda.

GEOS-Chem (http://www.geos-chem.org) is a global 3-D chemical transport model driven by meteorological fields from the Goddard Earth Observing System (GEOS) of the NASA Global Modeling and Assimilation Office (Bey et al., 2001; Eastham et al., 2014). It has been widely used to study trace gases and aerosols in the atmosphere. Here we use the model version 11-01 (http://wiki.seas.harvard.edu/geos-chem/index.php/GEOS-Chem_v11-01) driven by the MERRA-2 reanalysis (at 2.5° longitude by 2° latitude resolution) to simulate $^{222}$Rn and $^{210}$Pb. The model simulates the emission, transport (advection, convection, boundary layer mixing), deposition, and decay of the radionuclide tracers (Liu et al., 2001; Brattich et al., 2017; Yu et al., 2018; Zhang et al., 2021). As a function of latitude, longitude, and month, $^{222}$Rn emission uses a customized emission scenario that was built upon previous estimates and evaluated against global $^{222}$Rn surface observations and vertical profile measurements (Zhang et al., 2021). GEOS-Chem uses the TPCORE advection algorithm of Lin and Rood (1996), calculates convective transport using archived convective mass fluxes (Wu et al., 2007), and uses the non-local boundary-layer mixing scheme implemented by Lin and McElroy (2010). The wet deposition scheme follows that of Liu et al. (2001) and includes rainout (in-cloud scavenging) due to large-scale (stratiform and anvil) precipitation, scavenging in convective updrafts, and washout (below-cloud scavenging) by precipitation (Wang et al., 2011). A modification to the large-scale precipitation scavenging scheme is included to use spatiotemporally varying cloud water contents from MERRA-2 instead of a fixed constant value in the original model (Luo et al., 2019). Dry deposition is based on the resistance-in-series scheme of Wesely (1989).

### 3. Results and Discussion
### 3.1 Seasonal Profiles
### 3.1.1 Back-Trajectories

Our results in Fig. 1 show that the summer months (June-August, JJA) are distinct due to the Bermuda High promoting easterly winds at latitudes south of Bermuda that turn north and become southwesterly (~ parallel to U.S. East Coast) towards Bermuda. The Bermuda high pressure system and its associated anticyclonic circulation in the boundary layer have been reported to be strongest in April–September (Merrill, 1994; Moody et al., 1995). This high pressure system breaks down in other months in favor of strengthened extratropical subpolar low pressure, thus yielding more air influence from the northwest and west (Arimoto et al., 1995; Davis et al., 1997), which is clearly evident in the other three seasonal panels of Fig. 1 and most pronounced in the winter months (December-February, DJF). In their analysis of air mass history leading to rain events over Bermuda, Altieri et al. (2013) observed more influence from air originating over water in warmer months (April–September) and faster moving air masses originating over the continental U.S. primarily in the colder months of October–March. Moody and Galloway (1988) also showed that cool months (October–March) were marked by more transport from the U.S. East Coast. It can be deduced from Fig. 1 that based on the farther reaching source areas of the back-trajectories in colder months, and especially DJF, that air moves faster in the boreal winter. Finally, we note that Figs. S1-S2 show the same results as Fig. 1 but with ending altitudes of 500 m and 1

km over Bermuda; the sensitivity tests indicate the same general results and thus we continue the
discussion using results based on 100 m.

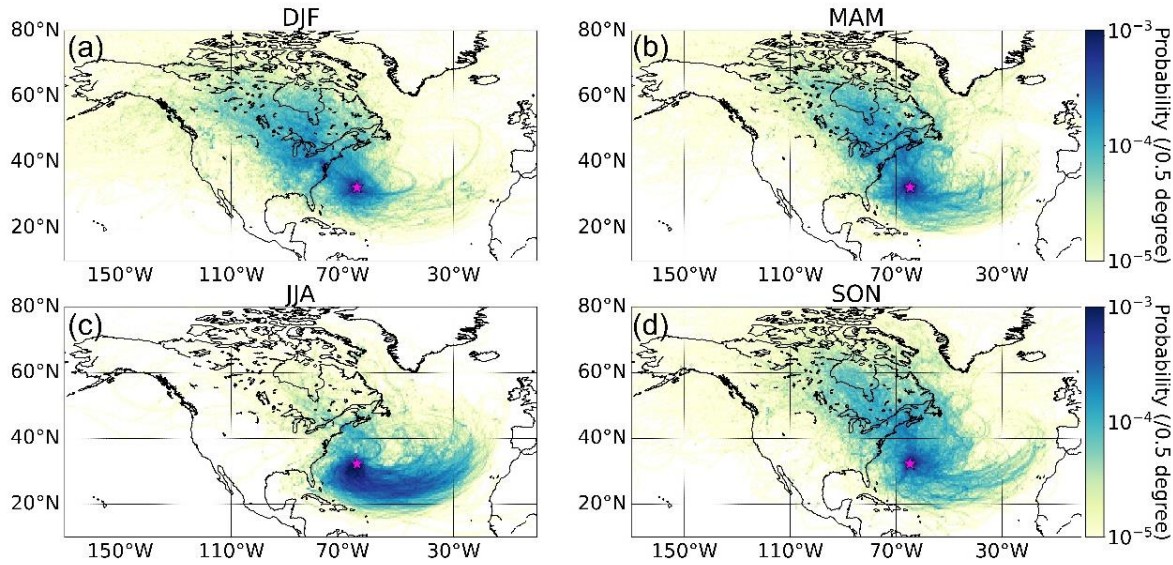

**Figure 1. Seasonal maps (a-d) showing the probability density of trajectories calculated**
**based on 10-day HYSPLIT backward trajectories reaching Bermuda (32.30° N, 64.77°W),**
**denoted by the pink star, at 100 m (AGL). This analysis is based on trajectories between 01**
**January 2015 and 31 December 2019. Analogous results for ending altitudes of 500 m and 1**
**km are shown in Figs. S1 and S2, respectively.**
**3.1.2 Surface Aerosol and NO$_x$**
Recent work has shown a seasonal cycle over Bermuda for column-integrated aerosol
properties, with aerosol optical depth (AOD) being highest in March-May (MAM) and JJA and
lowest in September-November (SON) and DJF (Aldhaif et al., 2021). They further showed that
sea salt contributed more to AOD in the colder months (SON, DJF) whereas sulfate, organic
carbon, black carbon and dust were more dominant in MAM and JJA. In their examination of
aerosol type seasonality at Bermuda, Huang et al. (1999) observed that marine and crustal elements
peaked in winter and summer, respectively, and that pollution-derived particles dominated in
spring with a smaller peak in fall. We use data from Fort Prospect station to gain a revised
perspective about seasonality and the weekly cycle of surface layer aerosol and additionally NO$_x$
(box notch plots in Figs. S3a-f).
Median seasonal concentrations of PM$_{2.5}$ (µg m$^{-3}$) were as follows at Bermuda, being
largely consistent with the AOD seasonal cycle: DJF = 5.50, MAM = 6.36, JJA = 6.11, SON =
5.33 (Fig. S3). NO$_x$ exhibited a similar seasonal pattern (ppbv): DJF = 17.76, MAM = 21.62, JJA
= 18.68, SON = 13.95 (Fig. S3). It is difficult to ascertain sources and impacts of precipitation on
PM$_{2.5}$ based on these values. As a next step we present the seasonal CWT maps showing the
predominant pathways accounting for the majority of PM$_{2.5}$ at Bermuda (Fig. 2). Expectedly, PM$_{2.5}$
in JJA is largely accounted for by trajectories following the general anticyclonic circulation already
shown in Fig. 1c associated with the Bermuda High. These air masses are enriched with African
dust as has been documented in many past studies (e.g., Arimoto, 2001; Huang et al., 1999; Muhs
et al., 2012). In contrast, the other seasons (especially DJF and MAM) showed greater relative
influence from North American outflow versus other source regions.

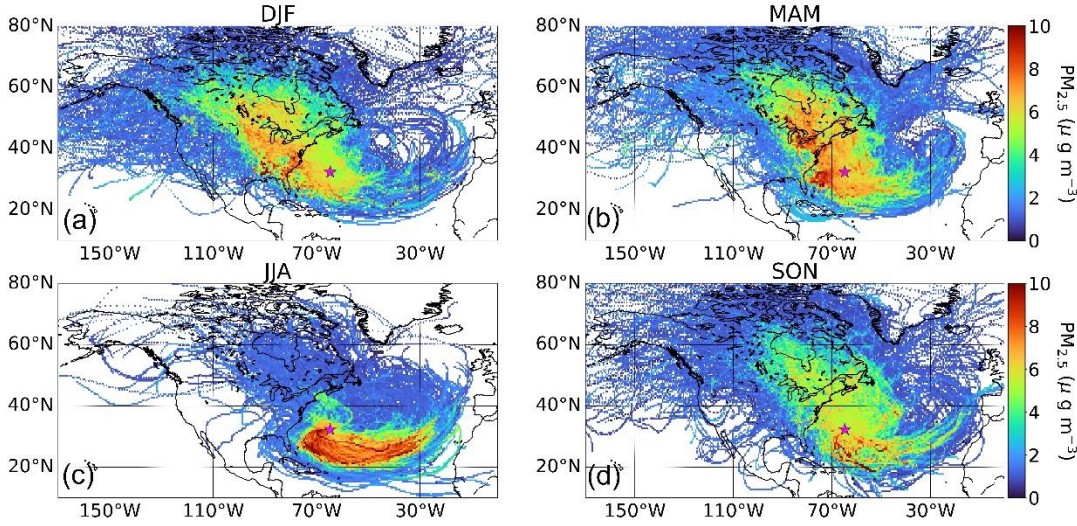

**Figure 2. Seasonal (a-d) concentration-weighted trajectory maps (CWT) for PM$_{2.5}$ measured**
**at Fort Prospect in Bermuda, denoted by the pink star. This analysis is based on trajectories**
**between 1 January 2015 and 31 December 2019.**
While we focus on long-range transport of PM$_{2.5}$ to Bermuda, local sources cannot be
ignored, including both sea salt and non-sea salt species (e.g., Galloway et al., 1988). The island
has a population of approximately 64,000 as of 2016 (Government of Bermuda, 2019). Local
influence from anthropogenic sources has been reported to be insignificant in contrast to
transported pollution (Galloway et al., 1988; Keene et al., 2014). We assess how significant local
anthropogenic sources are based on day-of-week aerosol concentrations and whether significantly
higher levels exist on working days as compared to weekend days as shown in other regions with
strong anthropogenic influence (Hilario et al., 2020 and references therein). Our analysis found
negligible difference between working days (Monday-Friday) and weekend days (Saturday-
Sunday) for both PM$_{2.5}$ and NO$_x$ when analysis was done based on annual (Figs. S3b/d) or seasonal
data (Figs. S4-S5). Therefore, it is less likely that local anthropogenic emissions dominate the
island's PM$_{2.5}$ and NO$_x$, providing support for transported sources being more influential; as will
be shown, normalizing PM$_{2.5}$ by CO helps control for local anthropogenic influence.
We also examined seasonal and day-of-week statistics for PM$_{10}$ to assess the relative
importance of coarse aerosol types including mainly sea salt and dust (Figs. S3e-f). Results reveal
the highest median PM$_{10}$ values (μg m$^{-3}$) in DJF (19.24), followed by MAM (18.51), JJA (17.98),
and SON (15.88). As will be shown later and already documented (Aldhaif et al., 2021), surface
wind speeds around Bermuda are highest in DJF, contributing to higher sea salt emissions.
Expectedly there was no observable PM$_{10}$ weekly cycle as dust and sea salt are naturally emitted.
Both PM$_{2.5}$ and PM$_{10}$ exhibited their highest seasonal standard deviations in JJA owing most likely
to the episodic nature of some pollution events such as with dust and biomass burning (e.g., Aldhaif
et al., 2021).

### 3.1.3 Precipitation Along Trajectories

Figure 3 shows seasonal profiles of average precipitation rate obtained from GDAS (Table 1) in 0.5°×0.5° grids based on 10-day back trajectories arriving at Bermuda (100 m AGL). The spatiotemporal pattern of precipitation over the WNAO is of most interest in terms of potential impacts on wet scavenging of aerosol during the transport of North American pollution to Bermuda. In that regard, DJF shows the most pronounced levels of precipitation to the north and northwest of Bermuda over the WNAO, coincident with strong and frequent convection linked to frontogenesis (Painemal et al., 2021). This is consistent with how Painemal et al. (2021) showed that precipitation exhibits maximum levels over the Gulf Stream path owing to relatively high sea surface temperature and strong surface turbulent fluxes.

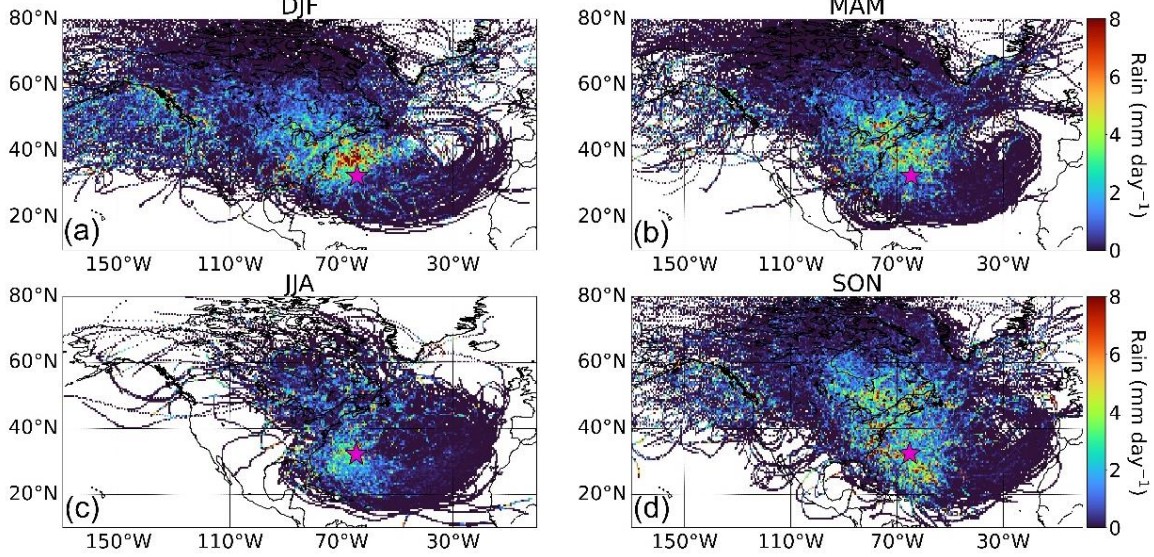

**Figure 3. Seasonal maps (a-d) of average precipitation occurring in 0.5°×0.5° grids based on 10-day backward trajectories reaching Bermuda (32.30° N, 64.77°W; pink star) at 100 m (AGL). This analysis is based on trajectories between 1 January 2015 and 31 December 2019.**

### 3.2 Trajectory Clustering

Prior to examining how precipitation directly impacts $PM_{2.5}$ at Bermuda, we identify characteristic trajectory pathways using the hierarchical agglomerative clustering method described in Section 2.3.2. We reiterate that this analysis is based on 4 days of back-trajectories, rather than 10 days from Figs. 1-3, to focus more on transport closer to Bermuda. The optimum solution based on the L-method (see Section 2.3.2) resulted in eight trajectory clusters (Fig. 4a), with five (numbered 1-5) coming from North America and the remaining three (numbered 6-8) more characteristic of the anticyclonic circulation described already for JJA. The former five clusters account for 49% of the total trajectories, with the latter three responsible for the remaining 51%. The majority of trajectories from North America come offshore north of North Carolina (i.e., coastal areas north of ~35°N) .

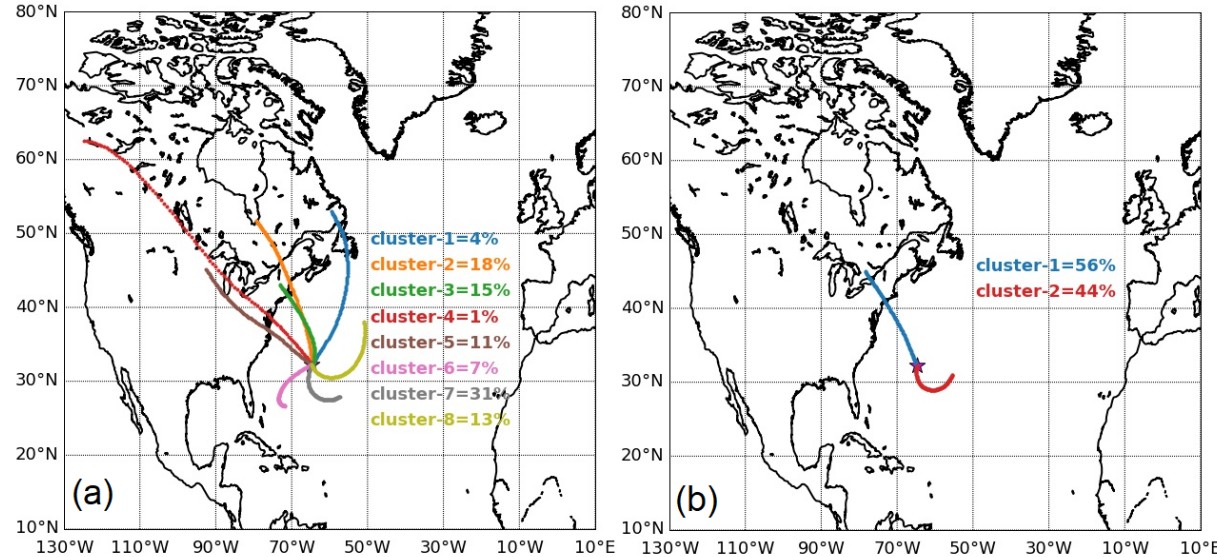

**Figure 4. Cluster mean trajectories based on the (a) optimum solution having eight clusters**
**and (b) a simplified solution with two clusters to enhance statistics for North American**
**trajectories. Clustering was performed on four-day HYSPLIT backward trajectories**
**between 1 January 2015 and 31 December 2019.**
For the sake of simplicity of the remainder of the discussion, we reduced the number of
characteristic trajectories to two (Fig. 4b), by conducting a new clustering analysis, to have one
from North America and the other from the southeast. Using only two clusters increases the
number of data points in the North American cluster for more robust calculations of rain-aerosol
relationships. Our choice to put together all North American air mass clusters into one group is
aligned with a similar clustering choice by Chen and Duce (1983; see their Fig. 3) where
trajectories were grouped together from Florida to the Canadian maritime provinces. Also, Mead
et al. (2013) divided trajectory data ending at Bermuda into "Saharan" and "non-Saharan" seasons
that generally coincide with our division of data into two clusters. Cluster 1 from North America
accounts for 56% of trajectories and Cluster 2 from the southeast is linked to 44% of trajectories.
It is clear from the two clusters that the North American air masses generally move faster as the
characteristic 4-day back-trajectories originate farther away from Bermuda than that of Cluster 2.
Regardless of season, Cluster 1 was associated with higher APT values with the seasonal
median values (units of mm) as follows (Cluster1/Cluster 2): DJF = 6.1/2.3; MAM = 5.2/1.8; JJA
= 6.7/2.8; SON = 7.0/5.1. Figure 5 shows a box notch plot comparing APT between clusters for
each season, demonstrating statistically significant differences in median values between clusters
for a given season at 95% confidence. Furthermore, Cluster 1 exhibited higher CO levels at
Bermuda for each season with median values (units of ppbv) as follows (Cluster 1/Cluster 2): DJF
=89.7/76.3; MAM = 88.5/75.0; JJA = 68.9/58.7; SON = 81.6/65.6. Therefore, the combination of
pollution outflow from North America and higher APT values makes Cluster 1 more relevant in
terms of identifying potential wet scavenging effects on transported aerosol over the WNAO. The
remainder of the study thus focuses on Cluster 1.

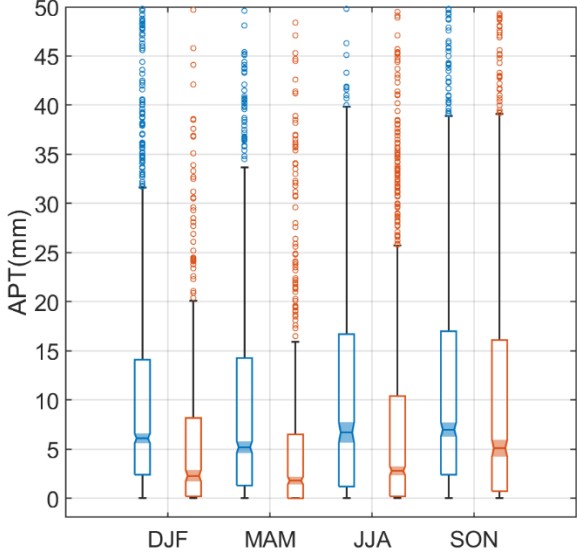

**Figure 5. Box notch plot for each season comparing accumulated precipitation along**
**trajectories (APT) for Clusters 1 (blue) and 2 (orange) from Fig. 4b. APT values were**
**estimated from four-day HYSPLIT back trajectories reaching Bermuda (32.30° N, 64.77°W)**
**at 100 m AGL. The middle, bottom, and top lines in each box represent the median, 25th**
**percentile, and 75th percentile, respectively. Markers show extreme values identified based**
**on 1.5×IQR (interquartile range) distance from the top of each box. Whiskers represent**
**maximum and minimum values excluding extreme points. Boxes with notches and shaded**
**regions that do not overlap have different medians at the 95% confidence level.**
**3.3 North America Trajectory Results**
We next examine the relationship between APT and aerosol transport to Bermuda based
on Cluster 1 results (Table 2). We compare data for "low" and "high" APT values based on
thresholds being the 25th percentile ($< 0.9$ mm) and 75th percentile ($> 13.5$ mm), respectively, based
on cumulative data from all seasons and years. As wet scavenging is expected to reduce $PM_{2.5}$
during its transport from North America to Bermuda, we anticipate lower $PM_{2.5}$ values at high
APT. However, the results indicate this is only the case for MAM and JJA, with similar median
values in SON and a higher median value in DJF for high APT conditions. Interestingly, NO, $NO_2$,
$NO_x$, and CO were all significantly higher in DJF for high APT conditions too. This raises the
issue that absolute $PM_{2.5}$ concentrations should be normalized to account for the differences in
concentration that existed closer to North America prior to potential wet scavenging over the
WNAO.
To study the effects of wet removal processes on aerosol particles during long-range
transport to a receptor site, many studies have used aerosol concentrations normalized by the
concentration of an inert gaseous species co-emitted with particles at distance sources. Such
normalization is critical and superior to the use of only aerosol concentration as the latter can be
influenced by local sources that can mask aerosol response to removal processes during long-range
transport. CO exhibits three important traits qualifying it as a species to normalize $PM_{2.5}$ by: (i) a
reliable marker of anthropogenic pollution stemming from North America (Corral et al., 2021);
(ii) being relatively insensitive to wet scavenging processes; and (iii) having a long lifetime in the
atmosphere (~1 month; Weinstock, 1969) compared to aerosol particles. Consequently, we
normalize $PM_{2.5}$ by $\Delta CO$ to quantify transport efficiency and to reveal the potential effects of wet
scavenging as has been done in past studies for other regions (Park et al., 2005; Garrett et al., 2010;
Hilario et al., 2021; Matsui et al., 2011; Moteki et al., 2012; Oshima et al., 2012). We first
determine the 5[th] percentile value of surface CO at Bermuda for each season for Cluster 1
trajectories and assume those are the seasonal background values as done also by Matsui et al.
(2011). We then calculate $\Delta CO$ as the difference between each 6-hourly CO data point at Bermuda
and the background value for a given season. We only use data when $\Delta CO > 3.2$ ppbv to ensure a
sufficiently high signal to noise ratio (Garrett et al., 2010).
**Table 2. Seasonal medians of aerosol, gas, and meteorological variables for Cluster 1 divided**
**into high- and low-APT categories. Differences in median values that are statistically**
**significant (p-value < 0.05) based on a Wilcoxon rank-sum test are highlighted with bold and**
**italic font. Percentage differences[*] between high- and low-APT median values are provided**
**in parentheses. NO, $NO_2$, $NO_x$, and $PM_{2.5}$ are based on Fort Prospect measurements, whereas**
**all other parameters are from MERRA-2 with the exception of the two APT rows (derived**
**from HYSPLIT and GDAS) and the last 8 rows corresponding to AERONET volume size**
**distribution data. We combined all seasons for AERONET data to have sufficient statistics**
**for comparisons (high APT = 16 points, low APT = 19 points). AERONET parameters**
**include volume concentration (V), effective radii ($R_{eff}$), volume median radii (R), and**
**geometric standard deviation (σ) with subscripts f and c for fine and coarse modes,**
**respectively. Number of data points for each table entry is summarized in Table S2.**


| Parameter | High-rain (APT > 13.5 mm)/Low-rain (APT < 0.9 mm) (% Difference[*]) | | | |
|---|---|---|---|---|
| | DJF | MAM | JJA | SON |
| NO (ppbv) | *6.0/3.5 (71 %)* | 7.3/7.8 (-6 %) | *8.3/13.1 (-37 %)* | 3.8/4.2 (-10 %) |
| $NO_2$ (ppbv) | *13.9/12.8 (9 %)* | 13.4/12.0 (12 %) | *8.6/6.6 (30 %)* | 9.4/9.2 (2 %) |
| $NO_x$ (ppbv) | *19.6/17.5 (12 %)* | 21.2/21.8 (-3 %) | *17.4/23.3 (-25 %)* | 14.1/14.2 (-1 %) |
| CO (ppbv) | *97.8/84.7 (15 %)* | *92.4/88.6 (4 %)* | *70.8/65.9 (7 %)* | 83.7/81.4 (3 %) |
| $PM_{2.5}$ ($\mu g\ m^{-3}$) | *6.1/5.5 (11 %)* | *6.7/7.3 (-8 %)* | *5.9/7.8 (-24 %)* | 5.5/5.1 (8 %) |
| $PM_{2.5}/\Delta CO$ ($\mu g\ m^{-3}\ ppbv^{-1}$) | *0.29/0.62 (-53 %)* | *0.35/0.51 (-31 %)* | 0.32/0.37 (-14 %) | 0.27/0.33 (-18 %) |
| Sea-Salt ($\mu g\ m^{-3}$) | *47.2/28.4 (66 %)* | *44.1/25.4 (74 %)* | 27.0/26.0 (4 %) | *50.6/36.0 (41 %)* |
| Sea-Salt$_{PM2.5}$ ($\mu g\ m^{-3}$) | *6.2/4.0 (55 %)* | *6.2/4.1 (51 %)* | 4.9/4.9 (0 %) | *6.8/5.0 (36 %)* |
| Dust ($\mu g\ m^{-3}$) | *0.80/0.91 (-12 %)* | *2.32/3.03 (-23 %)* | *4.47/3.02 (48 %)* | *1.16/1.04 (12 %)* |
| Dust$_{PM2.5}$ ($\mu g\ m^{-3}$) | *0.31/0.34 (-9 %)* | *0.79/1.00 (-21 %)* | *1.58/1.18 (34 %)* | *0.44/0.36 (22 %)* |
| Sea-Salt$/\Delta CO$ ($\mu g\ m^{-3}\ ppbv^{-1}$) | *2.10/2.74 (-23 %)* | *2.54/1.70 (49 %)* | 1.50/1.58 (-5 %) | *2.44/1.66 (47 %)* |
| Sulfate$/\Delta CO$ ($\mu g\ m^{-3}\ ppbv^{-1}$) | *0.029/0.055 (-47 %)* | *0.041/0.052 (-21 %)* | 0.039/0.046 (-15 %) | 0.024/0.027 (-11 %) |
| Dust$/\Delta CO$ ($\mu g\ m^{-3}\ ppbv^{-1}$) | *0.038/0.082 (-54 %)* | *0.129/0.186 (-31 %)* | *0.235/0.152 (55 %)* | 0.052/0.047 (11 %) |
| BC$/\Delta CO$ ($\mu g\ m^{-3}\ ppbv^{-1}$) | *0.0031/0.0056 (-45 %)* | *0.0042/0.0057 (-26 %)* | 0.0041/0.0049 (-16 %) | 0.0032/0.0033 (-3 %) |
| OC$/\Delta CO$ ($\mu g\ m^{-3}\ ppbv^{-1}$) | *0.0093/0.0238 (-61 %)* | *0.0164/0.0276 (-41 %)* | 0.0225/0.0287 (-22 %) | *0.0127/0.0153 (-17 %)* |
| Sea-Salt$_{PM2.5}/\Delta CO$ ($\mu g\ m^{-3}\ ppbv^{-1}$) | *0.263/0.403 (-35 %)* | *0.352/0.262 (34 %)* | 0.284/0.298 (-5 %) | *0.331/0.255 (30 %)* |
| Dust$_{PM2.5}/\Delta CO$ ($\mu g\ m^{-3}\ ppbv^{-1}$) | *0.015/0.033 (-55 %)* | *0.042/0.062 (-32 %)* | *0.087/0.053 (64 %)* | 0.018/0.017 (6 %) |
| Wind$_{SF}$ ($m\ s^{-1}$) | *8.5/7.1 (20 %)* | *8.4/5.9 (42 %)* | 4.4/4.7 (-6 %) | *7.7/6.6 (17 %)* |
| APT$_{6h}$ (mm) | *0.1/0.0 (NaN )* | 0.0/0.0 (NaN) | 0.0/0.0 (NaN) | 0.0/0.0 (NaN) |
| APT (mm) | *24.7/0.0 (NaN)* | *22.6/0.2 (11200 %)* | *24.1/0.0 (NaN)* | *25.0/0.2 (12400 %)* |

| | All |
|---|---|
| $V_f /\Delta CO \times 10^4$ ($\mu m^3\ \mu m^{-2}\ ppbv^{-1}$) | 3.42/7.55 (-55 %) |
| $R_{eff-f}$ ($\mu m$) | 0.158/0.147 (7 %) |
| $R_f$ ($\mu m$) | 0.176/0.171 (3 %) |
| $\sigma_f$ | 0.471/0.470 (0 %) |
| $V_c /\Delta CO \times 10^4$ ($\mu m^3\ \mu m^{-2}\ ppbv^{-1}$) | 2.04/2.12 (-4 %) |
| $R_{eff-c}$ ($\mu m$) | 1.956/2.085 (-6 %) |
| $R_c$ ($\mu m$) | 2.503/2.562 (-2 %) |
| $\sigma_c$ | *0.684/0.647 (6 %)* |


$^{*}\% \ difference = \dfrac{X_{High-rain} - X_{Low-rain}}{X_{Low-rain}} \times 100$

With the normalization technique, $PM_{2.5}/\Delta CO$ exhibits lower values in the high APT
category for each season as compared to low APT conditions (Fig. 6), with differences between
medians being statistically significant in DJF and MAM based on p-value < 0.05 with a Wilcoxon
rank-sum test (Table 2). The DJF season exhibits the greatest reduction of this ratio (by 53%) in
high APT conditions (0.29 µg m$^{-3}$ ppbv$^{-1}$ versus 0.62 µg m$^{-3}$ ppbv$^{-1}$ based on median values; Table
2). Therefore, these results suggest that it is plausible that wet scavenging has a marked impact on
surface $PM_{2.5}$ at a remote ocean site in the WNAO. This also helps support the speculation
proposed by Aldhaif et al. (2021) that wet scavenging can reconcile why, in particular for DJF, the
high density of trajectories coming from North America correlates with a reduction in fine
particulate pollution arriving at Bermuda as compared to other seasons. It is noteworthy that the
highest median value of $PM_{2.5}/\Delta CO$ was for the low APT category of DJF providing support for
how that season has both greater influence of aerosol transport from North America (when the
precipitation scavenging potential is reduced during low APT periods) and the greatest sensitivity
to the effects of precipitation over the WNAO owing to the widest range in this ratio's value
between high and low APT categories.

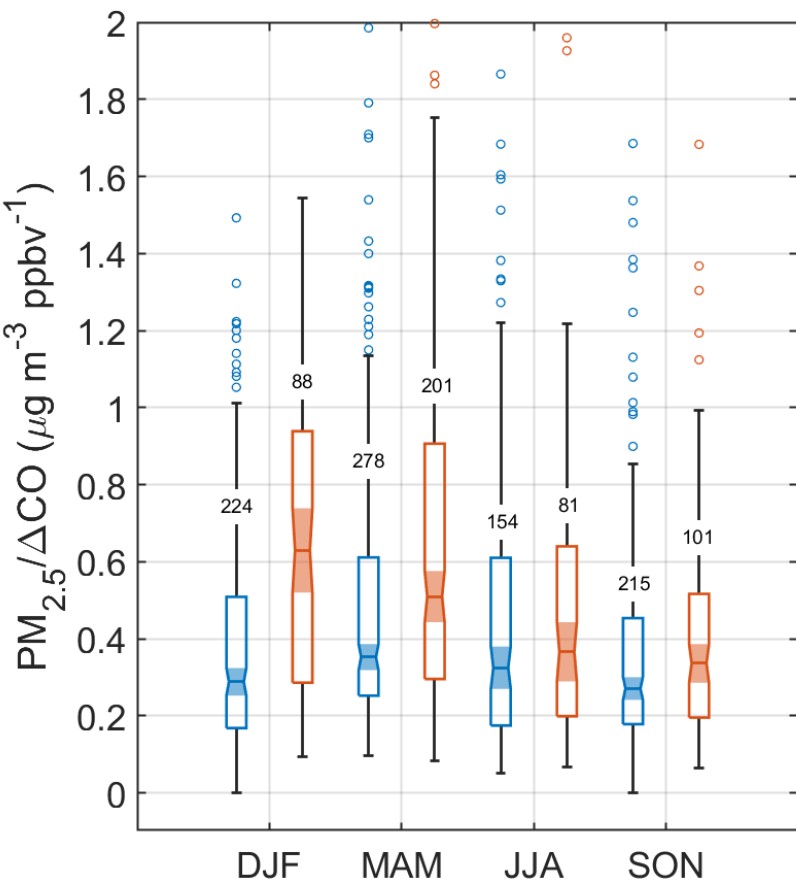

**Figure 6. Box notch plot for each season comparing the PM$_{2.5}$/ΔCO ratio for Cluster 1**
**trajectories for high-APT (blue) and low-APT (orange) conditions. APT thresholds are based**
**on 25$^{th}$ (< 0.9 mm) and 75$^{th}$ (> 13.5 mm) percentiles of APT for all trajectories reaching**
**Bermuda between 1 January 2015 and 31 December 2019. The number of samples in each**
**group is placed on whiskers.**
Figure 7 additionally shows the seasonal sensitivity of $PM_{2.5}/\Delta CO$ to APT based on four
bins of APT (bin ranges shown in Table S3) chosen in such a way to provide similar numbers of
data points per bin for each particular season. We note that the general trends are preserved using
similar bin ranges in each of the seasons. DJF and MAM show the greatest reductions from the
first to last bin as expected based on Table 2, but these also were the only two seasons showing
reductions between each successive bin. In contrast, SON and JJA exhibited more variable
behavior with $PM_{2.5}/\Delta CO$ actually increasing between a pair of bins in each season. A number of
reasons can potentially explain the less pronounced reduction in $PM_{2.5}/\Delta CO$ for SON and JJA: (i)
lower values to begin with in the lowest APT bins (and thus lower potential for scavenging to
occur); (ii) potential humidity effects associated with air masses at higher APT values promoting
secondary aerosol formation (Huang et al., 2014; Quan et al., 2015; Ding et al., 2021); (iii) more
influence from natural emissions in the form of dust (especially JJA) and sea salt (especially SON)
(Aldhaif et al., 2021). Another noteworthy result is that the season with the clearest scavenging
signature (DJF) shows the most sensitivity (i.e., steepest downward slope) between the first two
APT bins (0.9 mm versus 4.3 mm) as there was a 26% reduction in $PM_{2.5}/\Delta CO$ (0.584 $\mu g\ m^{-3}$
$ppbv^{-1}$ to 0.435 $\mu g\ m^{-3}\ ppbv^{-1}$), resulting in a slope (units of $\mu g\ m^{-3}\ ppbv^{-1}\ mm^{-1}$) of -0.044 in
contrast to slopes of -0.007 and -0.006 for the subsequent two pairs of bins in DJF. Tunved et al.
(2013) also reported a similar exponential trend between particle mass and accumulated
precipitation where an initial rapid decrease in particle mass was followed by a decreased removal
rate of aerosol due to precipitation.

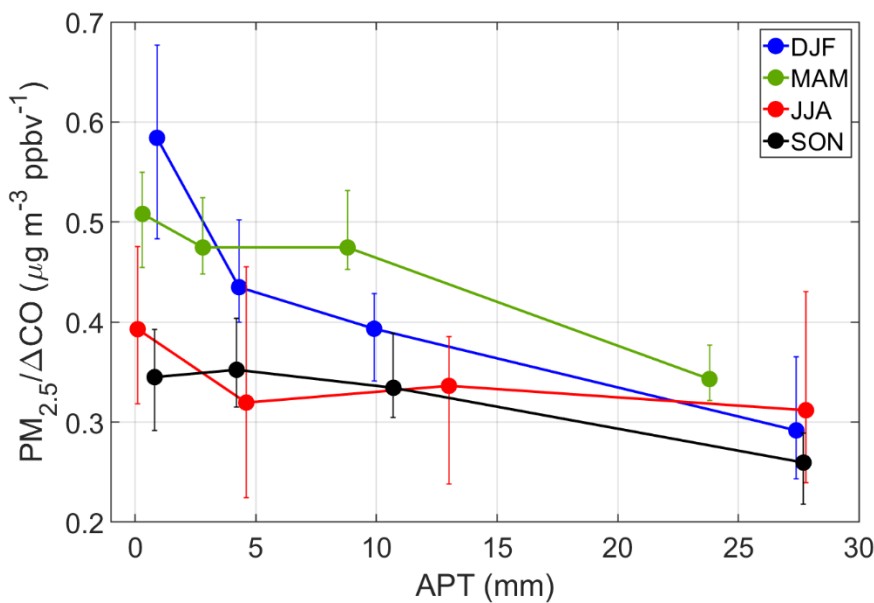

**Figure 7. Seasonal sensitivity of $PM_{2.5}/\Delta CO$ to APT for Cluster 1 trajectories, divided based**
**on four APT bins that have a similar number of data points per season. Markers denote**
**median values and error bars represent the 95% confidence interval for medians based on a**
**bootstrapping method (n = 100,000). Number of points per marker: DJF = 192 – 194; MAM**
**= 247 – 251; JJA = 107 – 110; SON = 183 – 191.**
We next address some additional details motivated by values shown in Table 2. We
examine three aerosol constituents linked to anthropogenic outflow from North America, including
sulfate, black carbon (BC), and organic carbon (OC) from MERRA-2 reanalysis. We recognize

that sulfate and OC have non-anthropogenic precursor vapors such as ocean-emitted dimethyl sulfide and biogenic volatile organic compounds, respectively. Being the most abundant of the three, sulfate exhibits the same characteristics as $PM_{2.5}$ when normalized by $\Delta CO$ with the sharpest reduction at high APT conditions in DJF, followed by MAM, and then finally by JJA and SON albeit with p-values > 0.05 for the latter two seasons as compared to low APT conditions. $BC/\Delta CO$ ratios show the same relative characteristics between APT categories as sulfate/$\Delta CO$ for each season, and mostly the same for OC/$\Delta CO$ except that the reduction in the median value in high APT conditions for SON was significant (p-value < 0.05). Regardless of season, but most pronounced in DJF, was the consistent result that OC/$\Delta CO$ exhibited the highest relative reduction at high APT conditions (versus low APT) compared to BC and sulfate. Further work with more expansive observational data is needed to better understand how different species respond to wet scavenging.

Normalization by $\Delta CO$ was important for assessing transport efficiency of anthropogenic pollution, but we also considered dust and sea salt without $\Delta CO$ normalization as they are predominantly emitted by natural sources. Although outside the scope of this study, we caution that MERRA-2 concentrations of sea salt in the $PM_{2.5}$ fraction may exceed those of total $PM_{2.5}$ as measured at Ft. Prospect (Table 2) owing to the inherent differences in the two respective datasets including the larger spatial scale covered by MERRA-2 as compared to the point measurements at Ft. Prospect. Previous analysis of precipitation scavenging ratios over Bermuda showed that larger aerosol types (e.g., sea salt) are removed more efficiently than smaller aerosol types (e.g., sulfate, nitrate) (Galloway et al., 1993). Total sea salt and sea salt in the $PM_{2.5}$ fraction exhibited higher median concentrations for the high APT category (p-value < 0.05) for all seasons except JJA, which had more comparable values. This can be explained by how days experiencing high APT exhibited significantly higher surface wind speeds around Bermuda for all seasons except JJA, for which wind speeds in general were depressed. Therefore, the reduction of the $PM_{2.5}/\Delta CO$ ratio in high APT conditions may actually be an underestimate of wet scavenging of North American pollution outflow since local sea salt is higher windier days marked by high APT.

To put this last assertion on firmer ground, we examined local rain values as they could be influential in terms of scavenging the locally generated sea salt. The median values of local rain on high APT days for each season based on APT for the most recent 6 hours of trajectories arriving at Bermuda ($APT_{6h}$) were $0.0 – 0.1$ mm, while median values of $APT_{6h}$ on low APT days were 0 mm in each season. The only significant difference in median $APT_{6h}$ values was in DJF when it was 0.1 mm on high APT days in contrast to 0.0 mm on low APT days. Therefore, for DJF the slightly enhanced $APT_{6h}$ can possibly offset the greater sea salt emissions in terms of impacting $PM_{2.5}$ levels over Bermuda. Results for the other major natural aerosol type (dust) reveal much lower overall concentrations as compared to sea salt for both bulk sizes and the $PM_{2.5}$ fraction. There was no consistent trend across the four seasons in terms of dust levels being higher for either the low or high APT category, which is not unexpected as dust is not a major aerosol type expected from North American outflow (Yu et al., 2020; Corral et al., 2021).

### 3.3.1 Volume Size Distributions

We next examine AERONET volume size distribution (VSD) relationships with APT. We normalize the volume concentration data by corresponding $\Delta CO$ in the same way as was done for $PM_{2.5}$, with the same condition of using data only when $\Delta CO > 3.2$ ppbv. A few cautionary details are first noted about these data in comparison to APT: (i) there are limited VSD data in the AERONET dataset, which is why we use all seasons of data together for Fig. 8 and Table 2; (ii)

AERONET data are representative of ambient conditions and changes in relative humidity can
influence VSD profiles; and (iii) AERONET data are column-based and not necessarily
representative of only the surface layer where the trajectories end in our analysis of HYSPLIT
data. Related to the last point, past work noted that column optical properties over Bermuda can
be weakly correlated with such measurements at the surface (Aryal et al., 2014) due largely to
aerosol layers aloft (Ennis and Sievering, 1990). At the same time, studies have shown that there
can be enhanced number and volume concentrations in the marine boundary layer versus the free
troposphere over Bermuda (Horvath et al., 1990; Kim et al., 1990).
The median VSDs for both APT categories exhibit a bimodal profile with a more dominant
coarse mode, consistent with what is already known for Bermuda based on AERONET data
(Aldhaif et al., 2021). The unique aspect of this work is that in high APT conditions, there is a
reduction in median volume concentration in the smaller mode between radii of 0.05 and ~1 µm,
with a slight enhancement on the leading shoulder of the larger mode between radii of 1.71 and
2.94 µm (Fig. 8). The greatest relative reductions in the fine mode, which is more indicative of
transported continental pollution, occurred between midpoint radii of 0.15 and 0.33 µm with
relative reductions in those four bins (i.e., midpoint radii = 0.15, 0.19, 0.26, and 0.33 µm) ranging
from 38% to 52%. The coarse mode peaked at larger radii (3.86 µm) in low APT conditions relative
to high APT conditions (2.94 µm).
Table 2 reports VSD parameter values for the APT categories separated by fine and coarse
modes. Although only significantly different based on 90% confidence (p-value = 0.09), the fine
mode volume concentration normalized by ΔCO in the high APT category was less than half (45%)
the value in the low APT category. There were insignificant differences between effective radii
and volume median radii, in addition to the geometric standard deviation for the fine mode between
APT categories. For the coarse mode, only the geometric standard deviation exhibited a significant
difference by being higher in the high APT category (0.684 versus 0.647), although we presume
that has less to do with actual scavenging effects and more to do with different times of the year
where the relative abundance of different coarse particle type changes.

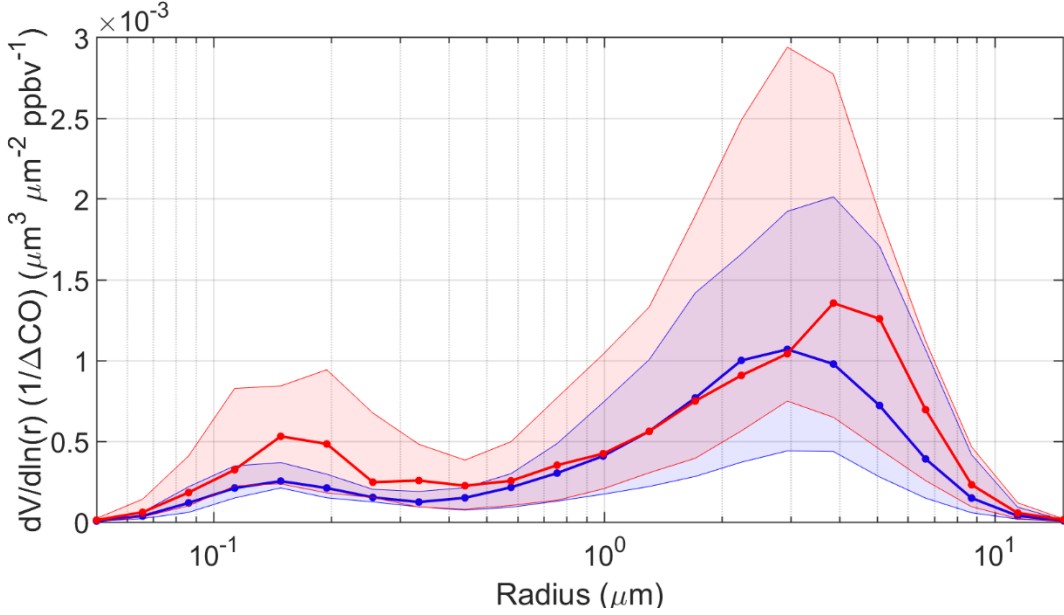

**Figure 8. Volume size distributions (VSD) normalized by ΔCO for high APT (> 13 mm; blue,**
**n = 16) and low APT (< 0.9 mm; red, n = 19) groups for Cluster 1 trajectories. Thick curves**

**correspond to medians and shaded areas extend to the 25th and 75th percentiles. VSDs are based on AERONET data between 1 January 2015 and 31 December 2019.**

The AERONET results support the idea that scavenging on high APT days efficiently removes fine particulate matter but that there can still be appreciable levels of locally generated sea salt due to higher local surface winds on high APT days. Related to the columnar nature of AERONET data, it is important to note that others have reported large-scale subsidence of pollution from the mid and upper troposphere, especially in spring, based on enhanced ozone mixing ratios at the surface of Bermuda (Oltmans and Levy, 1992; Cooper et al., 1998; Milne et al., 2000; Li et al., 2002). Moreover, this phenomenon is synoptically favorable with the transport of North American polluted air behind cold fronts especially in spring (Moody et al., 1995) and often linked to the lifting of polluted air out of the boundary layer by convection over the continental U.S. (Prados et al., 1999). It is unclear based on the current dataset how effective these events were in impacting either the surface layer or columnar-based aerosol measurements at Bermuda.

### 3.4 GEOS-Chem Model Results

We conduct four GEOS-Chem simulations of the $^{210}$Pb submicron aerosol tracer including a) one standard simulation; b) same as the standard simulation but with the $^{222}$Rn tracer emissions from the North American continent (25-60°N, 130-70°W) removed; c) same as the standard simulation but without $^{210}$Pb scavenging due to large-scale precipitation; and d) same as the standard simulation but without $^{210}$Pb scavenging by convective precipitation. The difference between a) and b) quantifies the North American contribution to atmospheric $^{210}$Pb concentrations. The difference between a) and c) reflects the role of large-scale precipitation scavenging, while the difference between a) and d) reflects that of convective precipitation scavenging in determining atmospheric $^{210}$Pb concentrations. All model simulations are conducted for the period from September 2016 to December 2017 with the first four months for spin-up. Monthly mean outputs for 2017 are used for analysis, which is a representative year within the time frame of the analysis presented in Sections 3.1-3.3. This is confirmed by the seasonal APT box chart constructed in Fig. S6 using only 2017 data, which nearly follows the trend observed when the five-year data are used (Fig. 5).

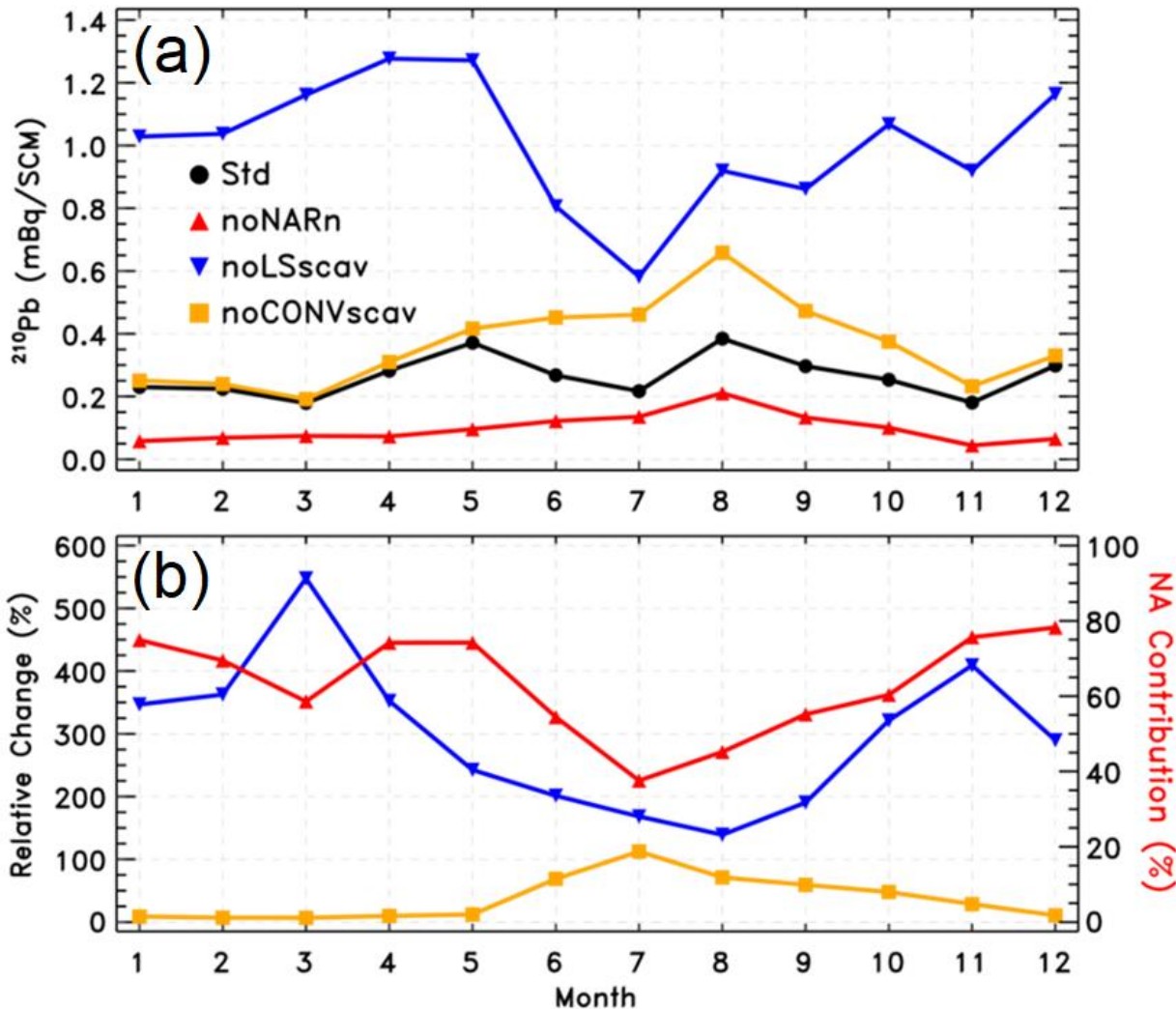

**Figure 9. Simulated monthly surface [210]Pb tracer concentrations submicron mBq/SCM at**
**Bermuda (32.31° N, 64.75° W) in 2017 as a way to assess effects of precipitation scavenging**
**on North American outflow. Panel (a): monthly mean surface [210]Pb concentrations in**
**the standard simulation ("Std") and three sensitivity simulations, i.e., without North**
**American [222]Rn emissions ("noNARn"), without large-scale precipitation scavenging**
**("noLSscav"), and without convective precipitation scavenging ("noCONVscav"). Panel (b):**
**percentage changes, i.e., (noLSscav-Std)/Std×100 in blue and (noCONVscav-Std)/Std×100 in**
**orange, and the North American contribution in red, i.e., (Std–noNARn)/Std×100.**
Figure 9a shows monthly mean surface [210]Pb concentrations at Bermuda for 2017 in
the standard simulation and three sensitivity simulations. Figure 9b plots the relative changes in
simulated [210]Pb concentrations due to the effects of large-scale or convective precipitation
scavenging. Also included in Fig. 9b is the North American contribution. The standard model
simulates a seasonality in [210]Pb concentrations with two distinct peaks in May and August (upper
panel). The May peak is a result of increased transport from North America in combination with
reduced scavenging. In contrast, the August peak results from long-range transport from other
continents (e.g., North Africa, Europe) along the southern edge of the Bermuda High. The lows in
March and November are attributed to strong large-scale precipitation scavenging, and the low in
July is associated with enhanced convective precipitation scavenging. The sensitivity simulations
clearly show that the role of large-scale precipitation scavenging in affecting surface $^{210}$Pb
concentrations at Bermuda is much larger in winter/spring than in summer, with a maximum in
March (lower panel), while convective scavenging also plays an important role in summer. The
relative contribution of North American $^{222}$Rn emissions is largest in winter (~75-80%), suggesting
air masses reaching Bermuda often experience large-scale precipitation scavenging while
traveling from the North American continent during winter. While the model may have limitations
and inherent uncertainties, its results are at least consistent with results shown already, putting our
conclusions on firmer ground.
**3.5 Airborne Case Study**
The DJF season has been shown in this study to exhibit the greatest potential for wet
scavenging and the highest density of trajectories from North America reaching Bermuda. To
probe deeper now, we take advantage of data from ACTIVATE RF6 on 22 February 2020, which
characterized the intermediate region between North America and Bermuda. Weather in the
ACTIVATE domain on this day was characterized by a transition from post-cold front conditions
to high pressure. A cold front passed over Bermuda the previous day at approximately 18:00 UTC
on 21 February, and by the flight period of RF06 was approximately 600 km southeast of the
island. Meanwhile, a broad but weakening area of surface high pressure continued eastward into
the southeast U. S. Winds in the boundary layer were southwesterly at around 5 m s$^{-1}$ near the base
of operations (NASA Langley Research Center; Hampton, Virginia), which were associated with
a weak trough on the northeast side of the high pressure system. These winds shifted to north-
northwest near the coast at 2.5 m s$^{-1}$ and north-northeast at 7.4 m s$^{-1}$ near the far end of the flight
track; Bermuda reported north-northeast winds around 9 m s$^{-1}$ during this period. Aloft, 500 hPa
flow was from the west-northwest. NASA Langley reported few to no clouds during the flight
period, while Bermuda reported broken clouds with multiple layers (with bases around 900 m and
1800 m) and rain showers at or near the airport. This is consistent with satellite imagery (Fig. 10a),
which shows an area of scattered to broken cumulus and stratocumulus extending from the cold
front near Bermuda to the edge of the Gulf Stream off the U.S. East Coast. Satellite-retrieved cloud
bases were at 1–2 km, with cloud tops ranging from 1.5–3.5 km; from the HU-25 Falcon flight
legs, cloud bases encountered along the flight track were 750–1100 m and cloud tops were 1200–
1800 m.

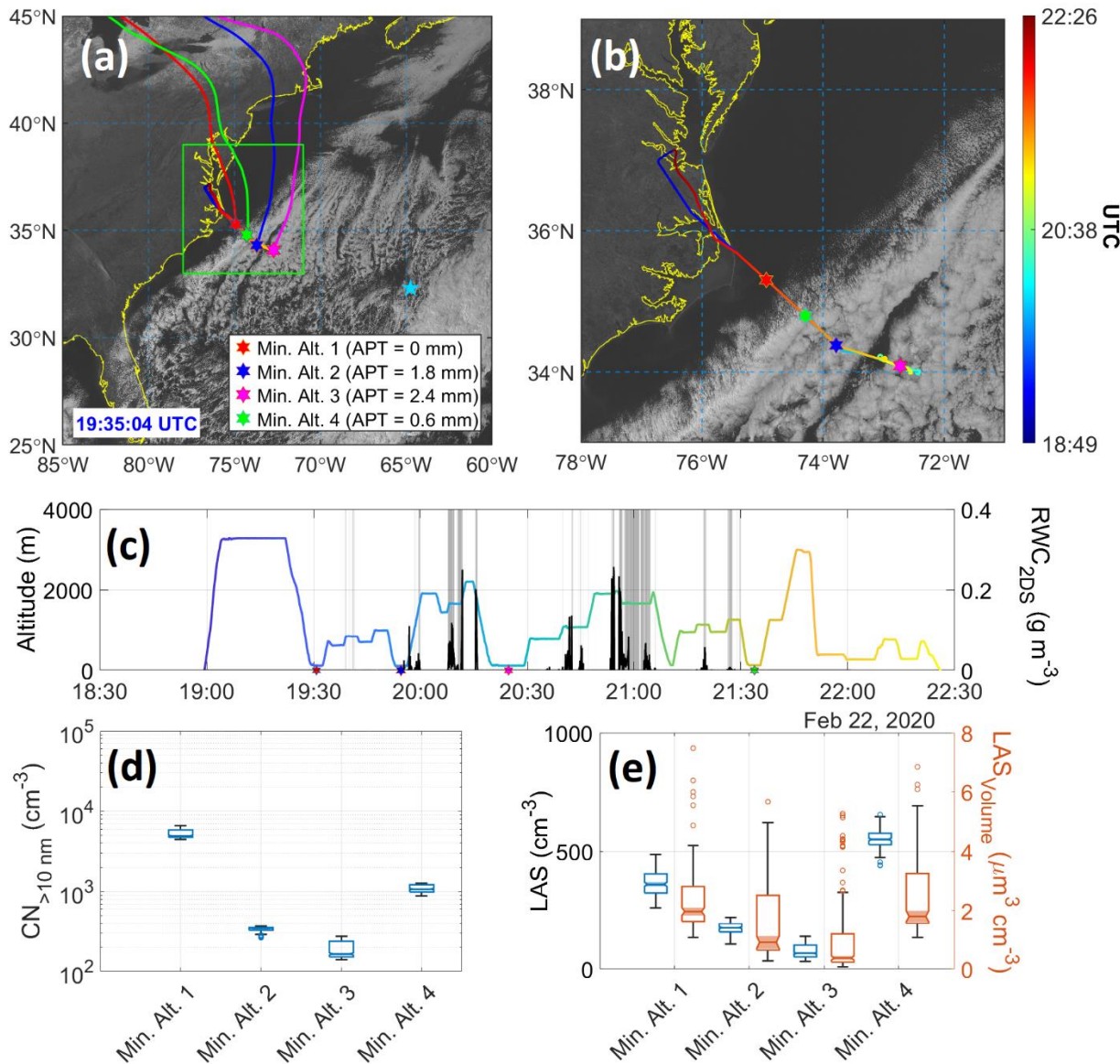

**Figure 10. Summary of ACTIVATE's Research Flight 6 on 22 February 2020. (a) HU-25 Falcon flight track overlaid on GOES-16 imagery of the WNAO (Bermuda denoted by blue star) also showing 96 hour back-trajectories calculated for each respective Min Alt. leg. The midpoint of the four Min. Alt. legs are marked including values for the accumulated precipitation along the trajectory (APT) for the recent history of the sampled air masses when they were over the ocean (time over land excluded from APT calculation). (b) Zoomed in version of panel (a) focused on the flight path. (c) Time series of Falcon altitude colored by flight UTC time (color bar in panel b) and rain water content (RWC) from the 2DS probe. Gray shaded bars signify when FCDP liquid water content exceeded 0.05 g m⁻³, indicative of cloud legs. The same four colored stars from (a) are shown on the x-axis to indicate where they occurred. (d-e) Box notch plots of the leg-mean Min. Alt. values of CPC particle concentration (> 0.01 μm), and the number and volume concentrations of the LAS (> 0.09 μm).**

Figure 10a shows the general flight path, which involved flying to a point southeast of the
operations base (Hampton, Virginia) and then re-tracing the path back to land. Four HYSPLIT
back-trajectories are shown (Fig. 10a) corresponding to midpoints of each Min. Alt. leg when the
aircraft was at its lowest altitude (~500 ft). APT calculations were conducted for segments of those
four trajectories that were over the ocean to focus on wet removal clouds over the WNAO.
Negligible rain accumulated up to the point of the Min. Alt. 1 leg, as there were cloud-free
conditions between land and that offshore point. In contrast, the next three Min. Alt. legs show
higher APT values ranging from 0.6 to 2.4 mm, consistent with the GOES-16 imagery showing
cloud fraction increasing just to the southeast of the Min. Alt. 1 leg. Expectedly, APT values
progressively increased with offshore distance as a result of air masses being exposed to clouds
for longer periods. Figure S7 shows 27 trajectories obtained for each Min. Alt. leg based on
ensemble trajectory analysis which is a technique available in HYSPLIT to evaluate uncertainties
in trajectory calculations by offsetting the meteorological data by a fixed grid factor. Average APT
values based on ensemble analysis (Fig. S7) were 0.29, 1.18, 2.27, and 0.73 mm corresponding to
Min. Alt. 1, 2, 3, and 4 legs, respectively, which follow the trend observed in Fig. 10.
Shortly after the Min. Alt. 1 leg, the Falcon conducted two consecutive pairs of BCB and
ACB legs (i.e., below cloud base followed by above cloud base), followed by a slant descent to
the Min. Alt. 2 leg, where RWC values were enhanced (up to 0.02 g m$^{-3}$ at 19:55:22 UTC) owing
to precipitation from overlying clouds. Very shortly thereafter, RWC reached as high as 0.11 g m$^{-3}$
$^{3}$ (19:56:50 UTC) in the slant ascent profile passing through clouds. The APT value in Min. Alt. 2
leg was 1.8 mm. A significant reduction was observed in the aerosol number and volume
concentrations for the Min. Alt. 2 leg as compared to the Min. Alt. 1 leg (Figs. 10d-e). Table S4
reports the statistics for aerosol parameters measured in Min. Alt. legs (Fig. 10). CPC (> 10 nm)
concentrations dropped by 93% from a leg-median value of 4938 cm$^{-3}$ during Min. Alt. 1 to 345
cm$^{-3}$ during Min. Alt. 2, whereas the LAS number and volume (> 100 nm) concentrations dropped
from 360 cm$^{-3}$ to 174 cm$^{-3}$ and from 2.0 µm$^3$ cm$^{-3}$ to 0.9 µm$^3$ cm$^{-3}$, respectively. Size distribution
data in those two legs show a significant reduction in particle concentration across the full diameter
range as measured by the SMPS and LAS (Fig. 11). A notable feature from the SMPS was a
pronounced peak between 3.5 – 14.1 nm suggestive of nucleation, that was absent in subsequent
Min. Alt. legs, presumably owing to some combination of coagulation and scavenging.

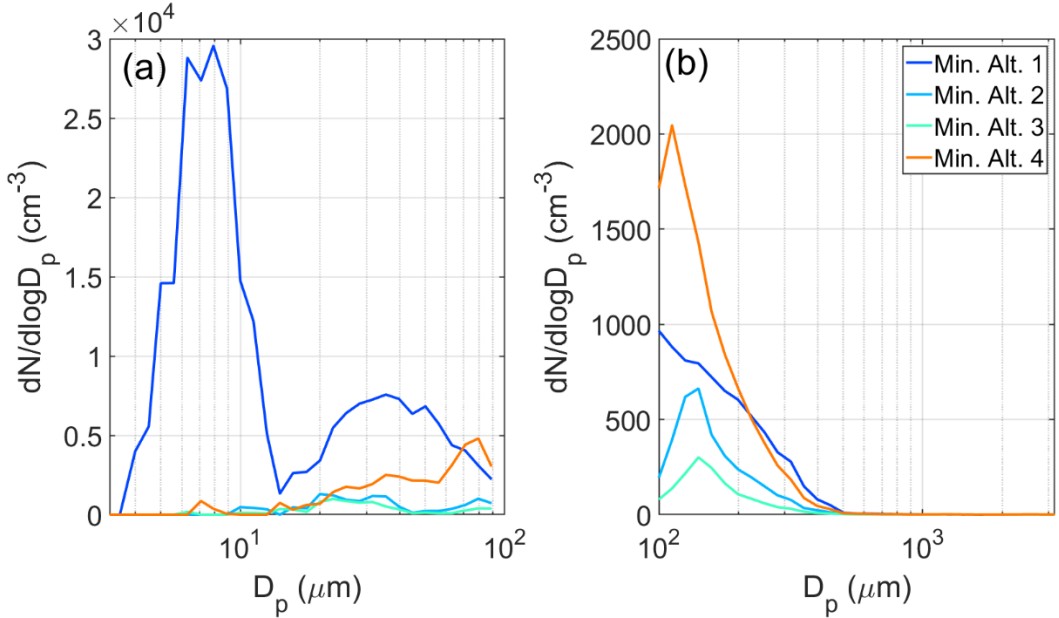


**Figure 11. Aerosol size distribution comparison (a = SMPS, b = LAS) between the four HU-25 Falcon Min. Alt. legs during ACTIVATE Research Flight 6, as shown in Fig. 10.**

The aircraft continued southeast after the Min. Alt. 2 leg and passed through more patches of precipitation, leading to the highest APT value of 2.4 mm in the Min. Alt. 3 leg, where leg-median values were as follows: CPC = 165 cm$^{-3}$, LAS number = 66 cm$^{-3}$, LAS volume = 0.4 μm$^3$ cm$^{-3}$. While the SMPS distributions in the Min. Alt. 2 and 3 legs were very similar, the LAS size distribution in the Min. Alt. 3 leg is shifted towards lower concentrations, especially below 400 nm. On the path back towards Virginia, the Falcon conducted one final Min. Alt. 4 leg right before the boundary between cloudy and clear air, with the APT value being 0.6 mm. Between the Min. Alt. 3 and 4 legs, again, significant RWC values were observed reaching as high as 0.26 g m$^{-3}$ at 20:54:20 UTC. Aerosol concentration measurements increased relative to the Min. Alt. 2 and 3 legs (leg-median values): CPC = 1076 cm$^{-3}$, LAS number = 545 cm$^{-3}$, LAS volume = 1.8 μm$^3$ cm$^{-3}$. It is difficult to compare results from the Min. Alt. 1 and 4 legs as ~2 hours had passed and there were different conditions impacting the two respective sampled air masses. The size distributions varied considerably for the Min. Alt. 4 leg as compared to the other three legs with increased concentrations between 20-200 nm, presumably as a result of continued pollution outflow and more photochemistry and aerosol growth processing as compared to earlier in the day.

To conclude, it is plausible based on the case flight data that the emerging presence of clouds and precipitation led to the substantial reduction of aerosol particles with distance offshore via wet scavenging processes. Further research is warranted with more extensive data to move closer to showing causal relationships between precipitation and aerosol particles. For instance, a few points of caution from RF6 are worth mentioning. First, the coastal trajectories in Fig. 10 corresponding to the different Min Alt. legs originated from varying places extending from the Virginia coast up north towards Cape Cod, Massachusetts. Secondly, cloud dynamics and boundary layer structure can vary offshore. Related to the latter, PBLH data obtained from MERRA-2 along the flight track revealed that there were deeper boundary layers farther offshore, but not sufficiently deeper to fully explain the reductions in aerosol concentration: PBLH

corresponding to Min. Alt. 1/2/3/4 = 1156/1728/1740/1530 m. Lastly, aerosol concentrations
linked to continental outflow naturally decrease anyways offshore, including in cloud-free
conditions, owing to dilution during transport.
**4. Conclusion**
This study examines the sensitivity of surface aerosol characteristics over a remote area of
the western North Atlantic Ocean (Bermuda) to precipitation along trajectories coming from North
America. Based on trajectory clustering with HYSPLIT data, two characteristic transport corridors
to Bermuda's surface layer (100 m AGL) were identified, with the focus being the one coming
from North America (Cluster 1). Seasonal analysis of HYSPLIT and Bermuda surface data showed
that JJA is distinct in terms of having transport from the southeast with the other seasons, especially
DJF, having more North American influence with higher concentrations of CO. Comparing Cluster
1 trajectories data between high (>13.5 mm) and low (<0.9 mm) accumulated precipitation along
trajectories (APT), there was a clear signature of wet scavenging effects by precipitation with more
than a two-fold reduction in $PM_{2.5}/\Delta CO$ in DJF (0.29 µg m$^{-3}$ ppbv$^{-1}$ versus 0.62 µg m$^{-3}$ ppbv$^{-1}$),
with the reduction being less severe for other seasons. The greatest sensitivity of $PM_{2.5}/\Delta CO$ to
APT was at the lowest values (up to ~5 mm; slope of -0.044 µg m$^{-3}$ ppbv$^{-1}$ mm$^{-1}$), above which
the descending slope of $PM_{2.5}/\Delta CO$ versus APT was less steep.
Speciated data indicate that anthropogenic species such as sulfate, black carbon, and
organic carbon are reduced as a function of APT (much like $PM_{2.5}$). However, sea salt was not
necessarily reduced and at times could even be higher at Bermuda with high APT conditions,
which is attributed to higher local wind speeds. Analysis of AERONET volume size distribution
data at Bermuda confirms the substantial reduction of fine mode volume concentrations in contrast
to a smaller change in the coarse mode on high APT days. GEOS-Chem simulations of the
radionuclide aerosol tracer $^{210}$Pb confirm that North American influence at the surface of Bermuda
is highest in DJF, with those air masses significantly impacted by large-scale (i.e., stratiform and
anvil) precipitation scavenging; furthermore, convective scavenging is shown to play an important
role in summer months. A research flight from ACTIVATE on 22 February 2020 demonstrates a
significant gradient in aerosol number and volume concentrations offshore of North America as
soon as trajectories start passing across clouds, consistent with increasing APT away from the
coast leading to increased aerosol particle removal.
Our results have implications for other remote marine regions impacted by transport of
continental emissions. These results also highlight the important role of precipitation in modifying
aerosol levels, including potentially their vertical distribution (e.g., Luan and Jaeglé, 2013), along
continental outflow trajectories. We show that cloud and precipitation processes along trajectories
have significant impacts on resultant aerosol characteristics. Therefore, it is important to strongly
constrain we scavenging processes in models to improve aerosol forecasting over the WNAO

*Data Availability.*
Fort Prospect Station Aerosol/Gas Measurements:
https://doi.org/10.6084/m9.figshare.13651454.v2
AERONET: https://aeronet.gsfc.nasa.gov/
HYSPLIT: https://www.ready.noaa.gov/HYSPLIT.php
MERRA-2: https://disc.gsfc.nasa.gov/
GEOS-Chem Model: http://wiki.seas.harvard.edu/geos-chem/index.php/GEOS-Chem_v11-01
Section 3.5 ACTIVATE Airborne Data:
https://doi.org/10.5067/SUBORBITAL/ACTIVATE/DATA001
Section 3.5 airport weather data: http://mesonet.agron.iastate.edu/ASOS/
Section 3.5 ocean surface analysis charts and GFS 500 hPa analysis:
https://www.ncei.noaa.gov/data/ncep-charts/access/
Section 3.5 North America Analysis/Satellite composite:
https://www.wpc.ncep.noaa.gov/archives/web_pages/sfc/sfc_archive_maps.php
Section 3.5 Satellite imagery/products: https://satcorps.larc.nasa.gov/cgi-
bin/site/showdoc?docid=4&cmd=field-experiment-homepage&exp=ACTIVATE
*Author contributions.* HD and MA conducted the analysis. AS and HD prepared the manuscript.
HL and BZ performed GEOS-Chem model radionuclide simulations and output analysis. All
authors contributed by providing input and/or participating in airborne data collection.
*Competing interests.* The authors declare that they have no conflict of interest.
*Acknowledgments.* The work was funded by NASA grant 80NSSC19K0442 in support of
ACTIVATE, a NASA Earth Venture Suborbital-3 (EVS-3) investigation funded by NASA's Earth
Science Division and managed through the Earth System Science Pathfinder Program Office. HL
and BZ acknowledge support from NASA grant 80NSSC19K0389. The AERONET station in
Bermuda is maintained on behalf of NASA by the Bermuda Institute of Ocean Sciences at the
Tudor Hill Marine Atmospheric Observatory, which is currently supported by NSF award
1829686, and by previous such awards during the time period in this study. Gas chemistry and PM
data at Fort Prospect are from the Bermuda Air Quality Program, operated by BIOS with funding
from the Department of Environment and Natural Resources, Government of Bermuda. The
authors acknowledge the NOAA Air Resources Laboratory (ARL) for the provision of the
HYSPLIT transport and dispersion model and READY website (http://ready.arl.noaa.gov) used in
this work. The Pacific Northwest National Laboratory (PNNL) is operated for DOE by the Battelle
Memorial Institute under contract DE-AC05-76RLO1830. The NASA Center for Climate
Simulation (NCCS) provided supercomputing resources. The GEOS-Chem model is managed by
the Atmospheric Chemistry Modeling Group at Harvard University with support from NASA
ACMAP and MAP programs. GEOS-Chem input files were obtained from the GEOS-Chem Data
Portal enabled by Compute Canada.

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
