# Peer review of "Aerosol Responses to Precipitation Along North American Air Trajectories Arriving at 2 Bermuda 3 4 Hossein Dadashazar1, Majid Alipanah2, Miguel Ricardo A. Hilario3, Ewan Crosbie4,5, Simon Kirschler6,7, Hongyu Liu8, Richard H. Moore<"

_Atmospheric Chemistry and Physics, 2021_

## Author Comment (AC2)

Author Response to Both Referee Comments:

Response: We thank the two reviewers for thoughtful suggestions and constructive criticism that have helped us improve our manuscript. Below we provide responses to reviewer concerns and suggestions in blue font.

**Reviewer 1:**

**Summary**

Hossein Dadashazar et al. investigated the impact of precipitation on aerosol particles along the airmass trajectories. Specifically, they investigated observed data from Bermuda locating in the North Atlantic Ocean east from the U.S East Coast. They studied how the mass concentration and volume size distributions of particulate matter is affected by the precipitation. They show that trajectories coming from North America during wintertime had the highest accumulated precipitation causing large reductions in the PM2.5/ΔCO (defined as "PM with aerodynamic diameter less than 2.5μm normalized by the enhancement of carbon monoxide above background"). They observed that changes in PM2.5/ΔCO were most sensitive to accumulated precipitation up to 5mm. In addition, their study was accompanied with GEOS-Chem model simulations providing information on wet scavenging versus convective scavenging, and a recent case study from aircraft field campaign ACTIVATE. Overall, the manuscript is very well written and structured, and the scientific methods are accurate. Studies concerning the scavenging of observed aerosols due to precipitation along airmass trajectories are sparse (Tunved et al., 2013, is the only one I am aware, and it has an arctic location), thus these types of studies are very much welcomed in our field. You could emphasize the sparseness of these studies (thus highlighting the need for yours) by referring to Tunved et al. I do recommend this study for publication, and it is well in the scope of ACP. I have listed some questions and suggestions below.

Response: Thanks for suggesting the work done by Tunved et al. (2013). We added the text below in the introduction:

"Overall, more work is warranted to better constrain wet scavenging of aerosol particles along trajectories as such studies are sparse not only for the WNAO but also for other regions (Tunved et al., 2013; Hilario et al., 2021)."

**Specific/minor comments:**

PM2.5 normalization with ΔCO: Could you elaborate a bit more why this kind of normalization is useful/more suitable instead of using the PM2.5 data as it is? You have given references to other studies using this technique in lines 406-417 where you describe what you have done, but a short sentence justifying the technique is missing.

Response: Normalizing $PM_{2.5}$ concentration with the concentration of a gas is a crucial step in studying the removal of aerosol during long-range transport as the aerosol concentration in a receptor site can be greatly influenced by local sources. Using an inert gas as a normalizing agent,

that gets co-emitted with aerosol at the distance sources, can reveal the level of scavenging aerosol particles in their path to the receptor site. We added the text below to address this comment:

"To study the effects of wet removal processes on aerosol particles during long-range transport to a receptor site, many studies have used aerosol concentrations normalized by the concentration of an inert gaseous species co-emitted with particles at distance sources. Such normalization is critical and superior to the use of only aerosol concentration as the latter can be influenced by local sources that can mask aerosol response to removal processes during long-range transport."

Table 1: In the caption you mention the GEOS-Chem data. This might be matter of opinion, but I would move those lines into the text as you have anyway only the observational data in the table.

Response: We removed this line from the caption as GEOS-Chem is already explained in the text.

I did not notice any information of the data coverage/missing data for the observations during the investigated time between 2015-2019. This information should be added to the supplement at least for readers to know e.g., if the representability of different seasons differs.

Response: We already provided the number of points used in the analyses conducted in the different parts of this paper (e.g., Figs. 6, 7, and 8 and also the current Table S2). The new Table S1 is additionally provided now to give the total number of points available for various measurements conducted at Bermuda. We also added the text below in Section 2.1 to mention major periods that the data were missing:

"There were a few periods when data were missing with the longest one being between 11 January 2016 and 08 April 2016 for the gases, and also between 16 October 2017 and 20 January 2018 for $PM_{2.5}$. There was no major discontinuity in $PM_{10}$ sampling. Table S1 reports the number of data points available for various seasons from the surface measurements at Fort Prospect in Bermuda.

Table S1. Seasonal number of points available for various measurements conducted at Fort Prospect in Bermuda between Jan 2015 and Dec 2019. It should be noted that data points correspond to 6-hour resolution for all variables except for $PM_{10}$ data that were at daily resolution.

|          | DJF  | MAM  | JJA  | SON  |
|----------|------|------|------|------|
| NO       | 1340 | 1626 | 1698 | 1758 |
| $NO_2$   | 1340 | 1626 | 1698 | 1758 |
| $NO_X$   | 1340 | 1626 | 1698 | 1758 |
| $PM_{2.5}$ | 1295 | 1761 | 1712 | 1551 |
| $PM_{10}$  | 57   | 66   | 72   | 59   |

Section 2.1: You average the data into 6-hourly values. Do you use mean, medians, or something else? This was not mentioned.

Response: We edited the corresponding line as follows:

"Hourly data were averaged over 6 hour intervals to match the time frequency of the trajectory data discussed subsequently."

Section 2.3: As you mention, you show first figures based on the 10-day back trajectories, but then limit the analysis to 4-day back trajectories. Why? I think some justification should be added. In Section 3.2 you shortly mention "to focus more on transport closer to Bermuda" but this still leaves an open question why to look those 10-day trajectories at all

Response: To address this comment, we added the following explanation to the end of section 2.3.1:

"10-day back trajectories were implemented for generating CWT and rain maps to illustrate potential distant sources impacting Bermuda. But for more quantitative analyses presented in the subsequent sections focused on transport most relevant to the WNAO region, four-day back trajectories were used by simply truncating 10-day trajectories. The use of four-day trajectories reduces the uncertainties associated with trajectory calculations in comparison to using 10-day trajectories and also enables us to focus on sources closer to the receptor site."

Line 230: there is a typo "GOES-Chem" which should read "GEOS-Chem"

Response: Fixed.

Figure 4 and related text: You combined the 8 clusters into 2 to enhance statistics. This is now related to the data coverage which was not mentioned. With your current trajectories you should have approximately 4(trajectories per day)x365(days per year)x5(2015-19) = 7300 which would be well enough statistics for 8 clusters. However, I assume the observational data is not complete which reduces the number of data rows to be used in the precipitation analysis? You mention in Table 2 caption the table in SI has number of data points for each table entry, but it does not give general picture of the available data as it has the high-and low APT categories.

Response: We added the total number of points for each season in Table S1. The reviewer is correct in their speculation that the observational data were not complete to allow for as robust of an analysis as we would prefer for the 8 clusters. Narrowing the analyses to two clusters make the discussion well-organized and easier to read. Thus, we decided to proceed with performing the analyses for only two trajectories.

Figure 5 and 6: As your y-axis ends at zero, the figure looks like the lower end of the boxes is cut out. Maybe show a bit of the negative axis too (we all know there is no negative rain) not to have this effect?

Response: We revised Figs. 5 and 6 according to this suggestion:

[Figure]

"

Figure 5. Box notch plot for each season comparing accumulated precipitation along trajectories (APT) for Clusters 1 (blue) and 2 (orange) from Fig. 4b. APT values were estimated from four-day HYSPLIT back trajectories reaching Bermuda (32.30° N, 64.77°W) at 100 m AGL. The middle, bottom, and top lines in each box represent the median, 25th percentile, and 75th percentile, respectively. Markers show extreme values identified based on 1.5×IQR (interquartile range) distance from the top of each box. Whiskers represent maximum and minimum values excluding extreme points. Boxes with notches and shaded regions that do not overlap have different medians at the 95% confidence level.

[Figure]

Figure 6. Box notch plot for each season comparing the $PM_{2.5}/\Delta CO$ ratio for Cluster 1 trajectories for high-APT (blue) and low-APT (orange) conditions. APT thresholds are based on 25th (< 0.9 mm) and 75th (> 13.5 mm) percentiles of APT for all trajectories reaching Bermuda between 1 January 2015 and 31 December 2019. The number of samples in each group is placed on whiskers."

Figures 6-8: It would be interesting to see how these differ for the cluster 2 as the airmasses have different characteristics, and as shown in Fig. 5, clearly lower APT too. These could be SI material.

Response: We did not perform such analyses for cluster 2 as we think CO is not a good normalizing factor for the cluster 2 trajectories as those are greatly impacted by natural sources and CO will not be as effective as we would prefer. Also, the main focus of this study is long-range transport for North American trajectories. Future work could look at trajectories coming from the direction of those in cluster 2.

Section 3.4: Why the GEOS-Chem simulations have only 2016-17 and not the same time period 2015-2019 as the observations? If I understand correctly, you did not do any "direct" comparisons (meaning having exactly the same time periods/simultaneous observations from GEOS-Chem and measurement data) between these datasets, is there a reason for that? Wouldn't

it be possible to output e.g., some aerosol concentrations from the model too? It seems so based on the description from section 2.5.

Response: We did not want to perform direct comparison between model outputs and the results from data analyses as this is beyond the scope of our work. The purpose of GEOS-Chem modeling was to shed light on the relative importance of different wet removal mechanisms impacting aerosol transport from North America to Bermuda, which we currently accomplish successfully.

Figure 10 and related text: Why is the time over land excluded here?

Response: Time over land is excluded as we wanted to focus on the wet removal that occurred by cloud systems over the ocean. We added the following lines to the text for further clarification:

"APT calculations were conducted for segments of those four trajectories that were over the ocean to focus on wet removal clouds over the WNAO."

Section 3.5: What is this "Min. Alt. Legs", was it explained somewhere?

Response: In the ACTIVATE mission, different legs are identified via specific names based on their altitude relative to the ocean surface and marine boundary layer clouds. We had already mentioned these legs and their altitude in section 2.4 as follows:

"The ACTIVATE strategy involves the HU-25 Falcon flying in the boundary layer to characterize gas, aerosol, cloud, and meteorological parameters along the following level legs: Min. Alt. = lowest altitude flown (500 ft), BCB = below cloud base, ACB = above cloud base, BCT = below cloud top, ACT = above cloud top."

Line 707-708: What are the numerical values reported here? I am not 100% sure after reading the text.

Response: We revised the text as follows:

"…aerosol concentration: PBLH corresponding to Min. Alt. 1/2/3/4 = 1156/1728/1740/1530 m…"

Line 720: State again here what the abbreviation "APT" stands for to make the conclusions more independent.

Response: we made the change as suggested:

"Comparing Cluster 1 trajectories data between high (>13.5 mm) and low (<0.9 mm) values of accumulated precipitation along trajectories (APT), …"

Line 732: What do you mean here with the "large-scale precipitation"?

Response: "Large scale precipitation scavenging" was explained in section 2.5, and was defined as precipitation caused by stratiform and anvil cloud systems. We edited this line for clarification:

"GEOS-Chem simulations of the radionuclide aerosol tracer [210]Pb confirm that North American influence at the surface of Bermuda is highest in DJF, with those air masses significantly impacted by large-scale (i.e., stratiform and anvil) precipitation scavenging; furthermore, convective scavenging is shown to play an important role in summer months."

**References**
Tunved, P., Ström, J., and Krejci, R.: Arctic aerosol life cycle: linking aerosol size distributions observed between 2000 and 2010 with air mass transport and precipitation at Zeppelin station, Ny-Ålesund, Svalbard, Atmos. Chem. Phys., 13, 3643-3660, 10.5194/acp-13-3643-2013, 2013.

**Reviewer 2:**

Hossein Dadashazar et al. investigated the impact of precipitation on aerosol particles along single particle back trajectories to Fort Prospect measurement station located on Bermuda. An assessment of the seasonal transport climatology to this station is provided by clustering the trajectories into different transport corridors. An investigation of the sensitivity of PM2.5 to experienced precipitation during transport is performed. This showed that air-masses arriving from North America during wintertime experienced the highest accumulated precipitation, which was found to correspond with large reductions in the PM2.5/ΔCO. These results are complemented and discussed with respect to GEOSChem model sensitivity simulations that provide insight into the role of large-scale precipitation scavenging versus convective wet scavenging, and a recent case study from a research flight from the ACTIVATE aircraft measurement campaign. Overall, the manuscript is very well written and structured. There are some minor instances where the readability can be improved, which I have indicated below. The scientific methods used in this study are accurate, although I do have some reservations about the applicability of single particle trajectories to assess transport during the ACTIVATE flight measurements. This study is well suited to the scope of ACP and I do recommend this study for publication. I have listed some questions and suggestions below to address prior to publication.

**General comments :-**
1.) The underlying results presented in this interesting study are based around combining in-situ observations with single particle trajectories calculated from the HYSPLIT trajectory model driven by GDAS reanalysis data. As with any study that is based around a model it is important to acknowledge any uncertainty in the results that are attributed to the modelling framework. Demonstration of the role of uncertainty in (a) the single-particle trajectory calculations and (b) the choice of reanalysis data, and their implications on the conclusions presented is currently missing and requires further attention.

(a) Uncertainties in the representation of air-mass history associated with single particle trajectories can be large. The accuracy of a single particle trajectory in representing the transport history compared to more detailed, e.g. dispersion modelling will depend on the prevalent

atmospheric conditions and the duration of the trajectories e.g. https://doi.org/10.5194/acp-9-8857-2009. Accordingly single-particle trajectory analysis is typically applied to investigate the average transport climatology to a receptor station by averaging many thousands of trajectories over long-time periods e.g. https://doi.org/10.5194/acp-13-3643-2013, and this approach is also applied in the majority of this study. However, for the comparison against the aircraft observations over a short-time period single-particle trajectories are also applied which will involve large uncertainties. As a minimum it would be appropriate for this section to run HYSPLIT in its trajectory ensemble configuration. The divergence of those trajectories would provide a more quantitative estimate of the uncertainty associated with the single particle trajectories currently shown.

Response: To address this comment regarding uncertainties associated with trajectory analysis, we added the new Figure S7 showing the results of ensemble trajectory analyses. As it's shown in Fig. S7, ensemble trajectories for research flight closely followed the ones shown in the main draft (Fig. 10), which are based on single trajectory calculations. We added the following lines to the draft to summarize the results shown in Fig. S7:

"Figure S7 shows 27 trajectories obtained for each Min. Alt. leg based on ensemble trajectory analysis, which is a technique available in HYSPLIT to evaluate uncertainties in trajectory calculations by offsetting the meteorological data by a fixed grid factor. Average APT values based on ensemble analysis (Fig. S7) were 0.29, 1.18, 2.27, and 0.73 mm corresponding to Min. Alt. 1, 2, 3, and 4 legs, respectively, which follow the trend observed in Fig. 10.

[Figure]

Figure S7. Trajectory ensembles for (a) Min. Alt. 1, (b) Min. Alt. 2, (c), Min. Alt. 3, and (d) Min. Alt. 4 legs, conducted by the HU-25 Falcon on 22 February 2020, overlaid on GOES 16 imagery obtained at 19:35:04 (UTC). The trajectory ensembles consisted of 27 individual trajectories obtained by offsetting the meteorological data by a fixed grid factor. Trajectory ensembles were calculated for the midpoints of four Min. Alt. legs, which are marked in each plot. The average (± standard deviation) of accumulated precipitation along the trajectory (APT) is also shown calculated for the recent history of the sampled air masses when they were over the ocean (time over land excluded from APT calculations). "

(b) There are numerous trajectory models available aside from HYSPLIT (e.g. Flexpart). For the calculation of single particle trajectories from these models I would expect the largest differences in the resulting trajectories to be due to the choice of meteorological (reanalysis) data used as input,

e.g.   https://doi.org/10.1080/10473289.2005.10464758;   https://doi.org/10.5194/asr-2-65-2008.
Currently there is no discussion of this uncertainty. The study uses GDAS reanalysis data to drive
HYSPLIT. HYSPLIT can be run using other reanalysis products, e.g. ERA-Interim, ERA5. How
would the results change if a different reanalysis product (e.g ERA-Interim) was used, with respect
to the air-mass history and experienced precipitation during transport? Significant differences in
precipitation       exist      between      different       reanalysis       products,       e.g.
https://doi.org/10.1002/2017RG000574;    https://doi.org/10.1016/j.jhydrol.2020.124632.    How
would the results change if you collocated precipitation from the Global Precipitation Climatology
Project (GPCP) along your GDAS trajectories?

Response:

We addressed this comment by adding a paragraph at the end of Section 2.3 explaining the
limitations of the methods implemented in this study. Comparing the results of using varying
precipitation data/meteorological datasets is out of the scope of this study and can be the subject
of future work:
        "Trajectory analyses contain errors that originate from factors including, but not limited to,
the choice of input meteorological data, resolution of input data, and the vertical transport method
used in trajectory calculations (Stohl et al., 1995; Cabello et al., 2008; Engström and Magnusson
2009). Although the choice of meteorological data is the most important contributor to the
uncertainties associated with trajectories calculations (Gebhart et al., 2005), no particular dataset
has been found to be superior in terms of yielding lowest error. While in this study we used GDAS
data, which have been widely used as input dataset for trajectory calculations even in regions with
complicated topography (e.g., Tunved et al., 2013; Su et al., 2015), the aforementioned inherent
errors should not be overlooked when interpreting the results presented in this work. Another factor
that can contribute to the uncertainties for the results presented in this work is the use of GDAS as
the source of precipitation data as previous works (Sun et al., 2018; Nogueira 2020) have
demonstrated that there is some level of disagreement between precipitation datasets."

2.) In the study the surface in-situ measurements are "averaged every 6 hours to match the time
frequency of the trajectory analysis". It is unclear why this is required and some clarification is
required as to why this approach was employed. With this type of trajectory modelling (single
particle trajectory calculations) there are no major limitations associated with computational cost
or data storage (very fast model, trajectory output data small). To my understanding the current
approach will result in the collocation of in-situ data averaged over 6 hours, to a trajectory arriving
at an instantaneous time. Are the measurements averaged such that the single trajectory arrives in
the middle of the 6hour time window? Why are hourly trajectories not calculated, and collocated
to the hourly measurements, as has been performed in previous similar studies, e.g.
https://doi.org/10.5194/acp-13-3643-2013. This would be advantageous as it would significantly
improve your statistical analysis (providing 24 trajectories per day as opposed to the current 4
trajectories per day), and more confidence that the trajectories represented the air-mass history
associated with the hourly in-situ measurements.

Response: We don't expect a significant change in the key results presented in this work using
hourly data. We did not use hourly data as we wanted to minimize the impacts of local sources and

diurnal cycles on the changes in PM$_{2.5}$ and gas concentrations. Furthermore, changes in long-range transport patterns normally occur over a daily timescale, thus we think trajectory calculations performed every 6 hours should be able to capture these changes. The same method was also implemented in a number of previous studies (e.g., Zhou et al., 2003; Kong et al., 2013; Dimitriou et al., 2015). We added the below lines to mention the benefit of averaging hourly data to 6-hour data:

"Hourly data were averaged over 6 hour intervals to match the time frequency of the trajectory data discussed subsequently. The conversion of hourly data to 6 hour data also helps to mask, to some extent, the unwanted effects of local sources and processes that occur on a small timescale. "

References:
https://doi.org/10.1016/j.atmosenv.2015.06.021
https://doi.org/10.1016/j.atmosenv.2003.12.034
https://doi.org/10.1016/j.atmosres.2012.10.012

3.) The study employs Modern-Era Retrospective analysis for Research and Applications- Version 2 (MERRA-2) as a data source for some aerosol parameters that are not measured in-situ at the station. How was this data connected to the HYSPLIT trajectories? The methodology (line 160) does not state whether this data was collocated along each of the individual trajectories prior to averaging. This would be beneficial; as it would also allow for a comparison of the reanalysis data to the in-situ measurements for certain key parameters to quantify the uncertainty associated with MERRA-2 as well as demonstrating how key parameters from MERRA-2 varied during transport, which would aid in understanding the role of precipitation on aerosol lifecycle during transport.

Response: We had already explained that MERRA-2 data were downloaded for a grid surrounding Bermuda which means they were not collected along trajectories but rather match temporally and spatially trajectories' arrival point at Bermuda. We added the following text for further clarification:

"Hourly and 3-hourly data were downloaded and averaged for a 0.5° latitude by 0.625° longitude grid (i.e., 32° – 32.5°N and 64.375° – 65°W) surrounding Bermuda and subsequently averaged over 6-hour intervals to match the time frequency of trajectory analysis results. It should be noted that MERRA-2 data were temporally and spatially coincident with the ending point of trajectories over Bermuda."

4.) As the key parameter used in the study is the accumulated precipitation along trajectories (APT) can the authors please provide some more details on how this was calculated/processed within the paper? Specifically, what does the precipitation diagnostic obtained from HYSPLIT-trajectories represent? Does it represent a precipitation rate that was converted to a total amount at each hour along the trajectory and then summed? Does it represent the precipitation at the height of each trajectory point, summed along the entire trajectory, or a column total precipitation at each point along a trajectory. If the latter, how did the authors process this data to ensure that the APT represents below cloud removal of aerosol? Some clarification is required here as if an individual trajectory is not at the height where the precipitation is occurring then this should be accounted for in the subsequent data processing.

Response: We used the former approach by integrating precipitation rates at the height of trajectories. We explained our method for APT calculations in Section 2.3 as follows:

"Trajectories were calculated using the GDAS one-degree archive data and with the "model vertical velocity" method, which means vertical motions were handled directly using meteorological data files. Moreover, accumulated precipitation along trajectories (APT) was calculated by integrating precipitation rates from GDAS, at the heights of trajectory endpoints, throughout the transport to the receptor site. "

5.) The study focusses on separating out transport corridors using a clustering method, however, the role of variation in vertical transport is not discussed. In figure 4 (b) what is the vertical transport climatology between cluster 1 and 2. What percentage of these clusters are dominated by low level transport more in which wet scavenging is more likely to play a role on aerosol removal? This should be considered in the analysis, and discussed. One solution would be to cluster the trajectories associated with cluster 1 and 2 with respect to height above the surface during transport. Furthermore, whilst the focus of the study is on the role of precipitation on aerosol during transport, the role of in-cloud scavenging is not discussed. What is the relative importance of in-cloud removal of aerosols during transport? This could be assessed by calculating the time spent in-cloud /out of cloud during transport to Bermuda. Such further analysis is required in order to support the final statement "wet scavenging processes in models require stronger constraints than other aerosol microphysical/chemical processes to improve the forecasting". Regardless, I recommend that this statement is rephrased slightly, to e.g. "it is important to strongly constrain we scavenging processes in models to improve the forecasting over the WNAO".

Response: The reviewer raised great points about the importance of vertical transport which was not the subject of current work. As reviewers suggested, future work is warranted to study the effects of vertical transport by applying clustering on trajectories in terms of height above the surface. We examined the relative role of different scavenging mechanisms by performing GEOS modeling; however, future works can focus on the role of in-cloud scavenging by using field measurements.

We followed reviewer's suggestion by rephrasing the line mentioned in this comment:
        "We show that cloud and precipitation processes along trajectories have significant impacts on resultant aerosol characteristics. Therefore, it is important to strongly constrain wet scavenging processes in models to improve aerosol forecasting over the WNAO "

6.) Key references that have performed similar analysis on the role of precipitation on the aerosol lifecycle are currently missing e.g. https://doi.org/10.5194/acp-13-3643-2013. Whilst performed in a different study region, such studies deserve inclusion in the discussion with respect to, for example, the interplay between precipitation and aerosol nucleation during the aerosol lifecycle.

Response: Thanks for suggesting this reference. We added such references to multiple places throughout the paper. Below is an example where we compare this other referenced study's results to the results shown in Fig. 7:

"Another noteworthy result is that the season with the clearest scavenging signature (DJF) shows the most sensitivity (i.e., steepest downward slope) between the first two APT bins (0.9 mm versus 4.3 mm) as there was a 26% reduction in PM2.5/$\Delta$CO (0.584 µg m-3 ppbv-1 to 0.435 µg m-3 ppbv-1), resulting in a slope (units of µg m-3 ppbv-1 mm-1) of -0.044 in contrast to slopes of -0.007 and -0.006 for the subsequent two pairs of bins in DJF. Tunved et al. (2013) also reported a similar exponential trend between particle mass and accumulated precipitation where an initial rapid decrease in particle mass was followed by a decreased removal rate of aerosol due to precipitation.

"

**Minor comments :-**

The study region is defined as a "ideal natural laboratory" within the same sentence that it is highlighted that it is a region that the region experiences pollution, i.e. not natural aerosol sources. A slightly rephrasing may be considered here.

Response: We aren't entirely sure what this comment was suggesting, but we did still change the word "ideal" to "suitable" in case that is a    ny better. We feel "ideal" was just fine, and that "suitable" should be fine in the context of this sentence.

Line 41: salt > sea salt

Response: Fixed

Line 65: "(Sorooshian et al., 2020), especially" > is the especially required in this context, simply remove?

Response: This line has been revised as follows:

"Consequently, Bermuda has been the subject of decades of intense atmospheric science research (Sorooshian et al., 2020), as it is a receptor site for both North African dust (Chen and Duce, 1983) and anthropogenic outflow from both North America"

Line 76: "along those trajectories": trajectories mentioned for the first time here in the introduction. Suggest introducing first, for example when first mention air-mass history can explain that this can be calculated using trajectory models.

Response: We introduced trajectories in the previous lines as follows:

"There have been extensive studies reporting on some aspect of air mass history, normally by calculating air parcel trajectories using transport and dispersion models, prior to arrival at Bermuda (Sorooshian et al., 2020 and references therein), including predominant circulation patterns impacting Bermuda at different times of the year (e.g., Miller and Harris, 1985; Veron et al., 1992)."

Line 77: Would benefit from some detail explaining why wet scavenging rates are difficult to constrain over the WNAO compared to other regions.

Response: We edited this line as follows:

"Wet scavenging rates are very difficult to constrain over open ocean areas such as the WNAO (Kadko and Prospero, 2011) not only because of complexity of physical mechanisms in play but also scarce necessary field measurements."

Line 82: "rain along trajectory pathways" > consider revising, e.g. precipitation transport history prior to arrival.

Response: Revised the line as suggested:

"their analysis of rain effects on nuclide activities were based on rain data collected at Bermuda without knowledge of precipitation transport history prior to arrival"

Line 83: Would benefit from an appropriate reference.

Response: We added references:

"While many studies have investigated how composition at Bermuda varies based on air mass trajectories (Miller and Harris, 1985; Cutter, 1993; Huang et al., 1996), the subject of how precipitation along those trajectories impact the resultant aerosol at Bermuda has not been adequately addressed but is motivated by past works (Moody and Galloway, 1988; Todd et al., 2003)."

Table 1: This table could be improved in places. The spatial resolution of certain datasets is missing. Is "Back-trajectory" a parameter? Consider splitting the table to provide one for in-situ observations, and one for model data.

Response: Table 1 has been updated based on the above suggestions. We removed back-trajectory from the table as a parameter. We also removed the website column as this information was already provided in the data availability section. Also, we decided to not divide the table into two parts as having all the parameters in one place is probably easier for reader. See below:

Table 1. Summary of datasets used in this work. Data are between 1 January 2015 and 31 December 2019, with the exception of ACTIVATE aircraft data based on a single flight day on 22 February 2020. Section 2 provides more details about the datasets used in this study, including specific instruments from the ACTIVATE airborne dataset.

| Parameter | Acronym | Data Source | Spatial Resolution | Time Resolution |
|---|---|---|---|---|
| Particulate matter mass concentration (aerodynamic diameter less than 2.5 µm) | $PM_{2.5}$ | Fort Prospect Station | - | Hourly |

| | | | | |
|---|---|---|---|---|
| Particulate matter mass concentration (aerodynamic diameter less than 10 µm) | $PM_{10}$ | Fort Prospect Station | - | Daily |
| Nitrogen monoxide concentration | NO | Fort Prospect Station | - | Hourly |
| Nitrogen dioxide concentration | $NO_2$ | Fort Prospect Station | - | Hourly |
| Nitrogen oxide concentration | $NO_X$ | Fort Prospect Station | - | Hourly |
| Volume size distribution | VSD | AERONET | - | Hourly |
| Carbon monoxide surface concentration | CO | MERRA-2 | $0.625° \times 0.5°$ | Hourly |
| Aerosol speciated surface mass concentrations | - | MERRA-2 | $0.625° \times 0.5°$ | Hourly |
| Surface wind speed | $Wind_{SF}$ | MERRA-2 | $0.625° \times 0.5°$ | Hourly |
| Planetary boundary layer height | PBLH | MERRA-2 | $0.625° \times 0.5°$ | Hourly |
| Precipitation | APT/Rain | GDAS | $1° \times 1°$ | Hourly |
| Aerosol/cloud properties | - | Airborne: ACTIVATE | - | 1 – 45 Sec |

Section 2.2: This section outlines the reanalysis data used within the study but does not mention the reanalysis data used in the trajectory modelling (GDAS).

Response: We moved the GDAS explanation to the section 2.2 as follows:

"The Global Data Assimilation System (GDAS) one-degree archive data were used for trajectory calculations explained in the subsequent section. Precipitation data were also obtained along the trajectories based on GDAS one-degree data."

Line 160: "converted to 6-hour data". Can you please clarify what is meant by converted? Averaged, or instantaneous fields?

Response: Revised as follows:

"Hourly and 3-hourly data were downloaded and averaged for a 0.5° latitude by 0.625° longitude grid (i.e., 32° – 32.5°N and 64.375° – 65°W) surrounding Bermuda and subsequently averaged over 6-hour intervals to match the time frequency of trajectory analysis results."

Line 165: "ending altitude of 100 m (AGL)". The height of the instrumentation AGL at the station is not provided for comparison. Due to the native resolution of the GDAS reanalysis data used for the trajectory calculations care needs to be taken for high altitude stations, as the average trajectory height at a location calculated by the trajectory model from 1-degree resolution reanalysis data may differ from the actual height.

Response: It is mentioned in section 2.1 that the measurement site at Fort Prospect in Bermuda was at 63 m ASL. Thus using 100 m AGL as the ending altitude is appropriate for our analyses.

We also ran some sensitivity tests with the results shown in the SI file that showed the transport paths remain similar using varying ending altitudes up to 1 km.

Line 171: "with the "model vertical velocity" method". Please consider rephrasing for clarity. I believe that this means that the vertical velocity is obtained from the underlying reanalysis data rather than being calculated from the HYSPLIT "model", see https://www.arl.noaa.gov/documents/workshop/NAQC2007/HTML_Docs/trajmetd.html

Response: Added further explanation as follows:

"Trajectories were calculated using the GDAS one-degree archive data and with the "model vertical velocity" method, which means vertical motions were handled directly using meteorological data files."

Line 175: "remaining sections of the paper are based on 4-day (96 hr) back-trajectories." I assume these are not from different trajectory calculations, but are the same trajectories simply truncated to end at 96hr instead of 240hr. Please clarify.

Response: Yes, the 6-hour trajectories were obtained by simply truncating 10-day trajectories. The following line has been added for clarification:

"10-day back trajectories were implemented for generating CWT and rain maps to illustrate potential distant sources impacting Bermuda. But for more quantitative analyses presented in the subsequent sections focused on transport most relevant to the WNAO region, four-day back trajectories were used by simply truncating 10-day trajectories."

Line 185: "A weight function following the method of Dimitriou et al. (2015) was applied". Please explain this method, demonstrating the equations used, for clarity.

Response: We provided details of weight function used:

"A weight function ($W_{ij}$ in Eq. 1) following the method of Dimitriou et al. (2015) was applied in the CWT analysis and precipitation maps to increase statistical stability. In Eq. 1, $n_{avg}$ is the average number of trajectory end points per individual gird cell over the study region excluding cells with zero trajectory points, while $n_{ij}$ is the number of trajectory end points in the grid cell (i,j)."

$$
W_{ij} =
\begin{cases}
1 & n_{ij} > 3\, n_{avg} \\[2em]
0.7 & 1.5\, n_{avg} < n_{ij} < 3\, n_{avg} \\[2em]
0.4 & n_{avg} < n_{ij} < 1.5\, n_{avg}
\end{cases}
\tag{1}
$$

| 0.2 | $n_{ij} < n_{avg}$ |

"

Line 224: 222Rn > define acronym - Radon 222 (222Rn)

Response: Revised:

"Lead-210 (210Pb, half-life 22.3 years) is the decay daughter of Radon-222 (222Rn, half-life 3.8 days) emitted mainly from land surfaces."

Line 269: "It can be deduced from Fig. 1 that based on the farther reaching source areas". The average wind speed can be easily calculated from the data in the trajectories, so there is no need to deduce from the figure. Pleases calculate instead.

Response: We did not think this level of detail is warranted as the trajectory distances are self-explanatory in terms of a qualitative link to transport speed.

Line 289: "Fort Prospect data to" > "data from Fort Prospect station to" Figures 2/3: One colour bar is sufficient for all subplots if the limits are constant for each.

Response: We revised the sentence as shown below. We think keeping a two color bar makes figures look nicer and does not impact the information being conveyed. Thus, we kept those figures as they were in original draft:

"We use data from Fort Prospect station to gain a revised perspective about seasonality and the weekly cycle of surface layer aerosol and additionally $NO_x$ (box notch plots in Figs. S3a-f)."

Line 317: Consider rephrasing to improve readability "including when resolved by season"

Response: Revised:

"Our analysis found negligible difference between working days (Monday-Friday) and weekend days (Saturday-Sunday) for both $PM_{2.5}$ and $NO_x$ when analysis was done based on annual (Figs. S3b/d) or seasonal data (Figs. S4-S5)."

Line 333: It is stated that 0.5 degree grids are used, however, to my understanding the GDAS data used is 1 degree. Why is the data averaged over a higher resolution than the underlying data source?

Response: We created CWT and precipitation maps by employing information at the trajectory endpoints rather than directly using GDAS data. It is true that GDAS data were at 1 degree resolution, but calculated trajectories were at higher resolution and basically they could be located anywhere in the study domain. As such, averaging trajectory data at higher resolution is acceptable as the same approach was used in previous papers (e.g., Kong et al., 2013; Masiol et al., 2017).

References:
https://doi.org/10.1016/j.atmosres.2012.10.012
https://doi.org/10.1016/j.atmosenv.2016.10.044

Figure 4: Please provide clarification on how Figure 4 (b) was created. On line 365 it states "Using only two clusters increases" which indicates that you have performed a new clustering, this time setting the number of clusters to 2 when you cluster. Or, does this mean you have simply grouped the clusters in Fig. 4a into two groups?

Response: No, a new clustering analysis was conducted with two clusters. We clarified this in the following lines:

"For the sake of simplicity of the remainder of the discussion, we reduced the number of characteristic trajectories to two (Fig. 4b), by conducting a new clustering analysis, to have one from North America and the other from the southeast."

Figure 4 (b). For clarity, would benefit from showing the trajectory members associated with each cluster in a different colour, e.g. light grey. This would help show the benefits of sub sampling the data using clustering over simply sectoring the transport pathways (e.g. select only trajectories that spend 95% time within certain lat/lon sectors).

Response: We admittedly did not understand this comment and what was being requested. We feel that Figure 4 and the associated text are quite clear already and that no changes are needed.

Line 382: Please consider rephrasing "more ideal" to improve readability.

Response: Revised as follows:

"Therefore, the combination of pollution outflow from North America and higher APT values makes Cluster 1 more relevant in terms of identifying potential wet scavenging effects on transported aerosol over the WNAO."

Line 382: "identifying potential wet scavenging effects". Some clarification is required on how it was determined that cluster 1 was more "ideal". Why was clustering of trajectories chosen as a method rather than sectoring? For example, an alternative strategy would be to bin the in-situ data by experienced precipitation during transport to identify the impact of wet removal for different transport corridors. This would provide valuable insight into the role of precipitation on the aerosol lifecycle for different aerosol regimes for this region.

Response: We feel as though the current text is sufficient to get our point across and that our methods and decision to end up with two final trajectory clusters were fine. All we were trying to say in this selected line is that Cluster 1 (from the northwest) has more influence from continental/urban emissions and that there is high APT, allowing for the signature of wet scavenging to be extracted from our data analysis. These characteristics were not nearly as strong for Cluster 2 from the southeast.

Line 403:405: This sentence is quite long and a little hard to digest. Please consider rephrasing to improve readability.

Response: We split this long sentence up into two sentences and had minor word changes:

"Interestingly, NO, $NO_2$, $NO_x$, and CO were all significantly higher in DJF for high APT conditions too. This raises the issue that absolute $PM_{2.5}$ concentrations should be normalized to account for the differences in concentration that existed closer to North America prior to potential wet scavenging over the WNAO."

Line 459: "four bins of APT chosen in such a way to provide similar numbers of data points per bin". Some clarification is needed here. What are the bin limits for each bin? Also, why is this approach required? Can the authors please demonstrate how the sensitivity in Figure 7 changes if a constant bin width is used, as performed in previous similar studies, for example Fig. 15: https://doi.org/10.5194/acp-13-3643-2013. A histogram showing the counts per bin can be provided, and a minimum number of counts for a bin to be considered can be implemented in such an approach. Statistics can easily be improved by using hourly trajectories (see general comment 2).

Response: We added a new Table S3 to give bin ranges that were used to create Fig.7:

"Figure 7 additionally shows the seasonal sensitivity of $PM_{2.5}/\Delta CO$ to APT based on four bins of APT (bin ranges shown in Table S3) chosen in such a way to provide similar numbers of data points per bin for each particular season. We note that the general trends are preserved using similar bin ranges in each of the seasons.

Table S3. Four APT bin ranges (mm) that were used to create seasonal plots shown in Fig. 7.

| Bin Number | DJF | MAM | JJA | SON |
|---|---|---|---|---|
| 1 | 0-2.5 | 0-1.4 | 0-1.8 | 0-2.4 |
| 2 | 2.5-6.7 | 1.4-5.3 | 1.8-8.7 | 2.4-7.2 |
| 3 | 6.7-15.7 | 5.3-14.6 | 8.7-19.0 | 7.2-16.0 |
| 4 | 15.7-164.9 | 14.6-118.9 | 19.0-74.2 | 16.0-106.2 |

We also ran a sensitivity test to show that the general trends observed in Fig. 7 remain the same if the same bin ranges were used in each season. The figure below shows the results when the following bin ranges were used for the four seasons: [0-2.5,2.5-6.7,6.7-15.7,15.7-164.9]:

[Figure]

Line 506: "This can be explained by how the high APT days" > consider rewording, e.g. "how days experiencing high APT…"

Response: Revised:

"This can be explained by how days experiencing high APT exhibited significantly higher surface wind speeds around Bermuda for all seasons except JJA, for which wind speeds in general were depressed."

Line 587: "2017 are used for analysis, which is a representative year". The authors should demonstrate in the SI, how it was ascertained that 2017 is a representative year by, for example, showing how the average precipitation/meteorology for this year compared to the average during 2015-2019.

Response: We added a new Figure S7 and the following text to verify this:

"Monthly mean outputs for 2017 are used for analysis, which is a representative year within the time frame of the analysis presented in Sections 3.1-3.3. This is confirmed by the seasonal APT box chart constructed in Fig. S6 using only 2017 data, which nearly follows the trend observed when the five-year data are used (Fig. 5)."

[Figure]

Figure S6. Box notch plot for each season of 2017 comparing accumulated precipitation along trajectories (APT) for Clusters 1 (blue) and 2 (orange) from Fig. 4b. APT values were estimated from four-day HYSPLIT back trajectories reaching Bermuda (32.30° N, 64.77°W) at 100 m AGL. The middle, bottom, and top lines in each box represent the median, 25th percentile, and 75th percentile, respectively. Markers show extreme values identified based on 1.5×IQR (interquartile range) distance from the top of each box. Whiskers represent maximum and minimum values excluding extreme points. Boxes with notches and shaded regions that do not overlap have different medians at the 95% confidence level."

Line 552: Could the authors please clarify how the mode statistics were calculated. I believe the correct procedure to obtain the average modal parameters provided in the table would be to calculate the modal parameters for each individual VSD, and then average these to obtain the average modal parameters, rather than calculating average modal parameters from the average VSD. Which approach was used?

Response: Correct, we took the former approach to estimate VSD statistics presented in Table 2.

Line 608:611: Could the authors please add some caveat to make it clear that these results pertain to a model for which the representation of aerosol – cloud – interactions are highly uncertain, and therefore, do not necessarily represent accurately what is happening in nature.

Response: Done:

"While the model may have limitations and inherent uncertainties, its results are at least consistent with results shown already, putting our conclusions on firmer ground."

Line 618: "this study to unite the greatest potential for wet scavenging" please consider rephrasing to improve readability.

Response: We changed the word "unite" to "exhibit", which should hopefully be more readable.

Line 658: "As a successful validation of the technique, no rain accumulated…" please consider rephrasing to improve readability.

Response: Edited:

"Negligible rain accumulated up to the point of the Min. Alt. 1 leg, as there were cloud-free conditions between land and that offshore point."

Line 656: "Four HYSPLIT back-trajectories are shown (Fig. 10a)". Please consider making this plot larger and giving it its own subplot label.

Response: We modified the figure and updated the caption as follows. We decided to address this comment by making the satellite image larger and changing the arrangement of other subplots without splitting the figure into two parts:

[Figure]

[Figure]

Figure 10. Summary of ACTIVATE's Research Flight 6 on 22 February 2020. (a) HU-25 Falcon flight track overlaid on GOES-16 imagery of the WNAO (Bermuda denoted by blue star) also showing 96 hour back-trajectories calculated for each respective Min Alt. leg. The midpoint of the four Min. Alt. legs are marked including values for the accumulated precipitation along the trajectory (APT) for the recent history of the sampled air masses when they were over the ocean (time over land excluded from APT calculation). (b) Zoomed in version of panel (a) focused on the flight path. (c) Time series of Falcon altitude colored by flight UTC time (color bar in panel b) and rain water content (RWC) from the 2DS probe. Gray shaded bars signify when FCDP liquid water content exceeded 0.05 g m-3, indicative of cloud legs. The same four colored stars from (a) are shown on the x-axis to indicate where they occurred. (d-e) Box notch plots of the leg-mean

Min. Alt. values of CPC particle concentration (> 0.01 µm), and the number and volume concentrations of the LAS (> 0.09 µm).

"

Line 658: Sentence beginning "As a successful validation …" please consider rephrasing to improve readability.

Response: We addressed this same comment above.

Page 23: Consider placing key statistics from the ACTIVATE research flight into a table to improve accessibility.

Response: We added a new Table S4 to summarize these statistics:

"A significant reduction was observed in the aerosol number and volume concentrations for the Min. Alt. 2 leg as compared to the Min. Alt. 1 leg (Figs. 10c-d). Table S4 reports the statistics for aerosol parameters measured in Min. Alt. legs (Fig. 10). CPC (> 10 nm) concentrations dropped by 93% from a leg-median value of 4938 cm$^{-3}$ during Min. Alt. 1 to 345 cm$^{-3}$ during Min. Alt. 2…."

**Table S4. Median values of aerosol parameters and APT for Min. Alt. legs (Fig. 10) conducted in ACTIVATE's Research Flight 6 on 22 February 2020.**

| Parameter | Min. Alt. 1 | Min. Alt. 2 | Min. Alt. 3 | Min. Alt. 4 |
|---|---|---|---|---|
| APT (mm) | 0.0 | 1.8 | 2.4 | 0.6 |
| $CN_{>10nm}$ (cm$^{-3}$) | 4938 | 345 | 165 | 1076 |
| LAS (cm$^{-3}$) | 360 | 174 | 66 | 550 |
| $LAS_{Volume}$ (µm$^3$ cm$^{-3}$) | 2.0 | 0.9 | 0.4 | 1.8 |

"

Line 729: "surface on days simultaneous with high APT trajectories." please consider rephrasing to improve readability.

Response: Edited:

"However, sea salt was not necessarily reduced and at times could even be higher at Bermuda with high APT conditions, which is attributed to higher local wind speeds."

Line 731: "in contrast to less change" > "to a smaller change in.."

Response: Revised as suggested.

---

## Author Response (AR2)

Author Response to Editor Comments:

Response: We thank the editor for these comments. We made the following changes in the manuscript:

Section 2.3.2: You use Haversine formula to calculate the distance matrix needed for the clustering. As you say, Haversine distance is calculated between two points in Earth. Are your final distances then based only to the endpoints of the trajectories or do you somehow average over the whole trajectory i.e., how is your final distance matrix constructed?

Response:

Distances between trajectories were calculated using the Haversine formula, which calculates distance between two points on Earth assuming they are on a great circle (Sinnott, 1984). The distance between any two trajectories was calculated as the sum of distances between trajectory endpoints. Subsequently, clustering was conducted based on the symmetric distance matrix, which includes the distances between all pairs of trajectories.

4.) As the key parameter used in the study is the accumulated precipitation along trajectories (APT) can the authors please provide some more details on how this was calculated/processed within the paper? Specifically, what does the precipitation diagnostic obtained from HYSPLIT-trajectories represent? Does it represent a precipitation rate that was converted to a total amount at each hour along the trajectory and then summed? Does it represent the precipitation at the height of each trajectory point, summed along the entire trajectory, or a column total precipitation at each point along a trajectory. If the latter, how did the authors process this data to ensure that the APT represents below cloud removal of aerosol? Some clarification is required here as if an individual trajectory is not at the height where the precipitation is occurring then this should be accounted for in the subsequent data processing.

Response: Precipitation data from GDAS are at the surface level thus APT values are the maximum potential level of wet removal experienced by trajectories. We explained our method for APT calculations in Section 2.3 as follows:

"Moreover, accumulated precipitation along trajectories (APT) was calculated by integrating surface precipitation data from GDAS throughout the transport to the receptor site. As GDAS precipitation data corresponds to the surface level, it should be noted that APT values presented in this study are associated with the potential maximum level of precipitation experienced by the air parcel through its transport journey. "